



# glmGUI v1.0: an R-based Geographical User Interface and toolbox for GLM (General Lake Model) simulations

Thomas Bueche [12], Marko Wenk[3], Benjamin Poschlod [1], Filippo Giadrossich [4], Mario Pirastru [4], Mark Vetter [5]

[1]Department of Geography, Ludwig-Maximilians-University Munich, Germany
[2]WipflerPLAN Planungsgesellschaft mbH, Pfaffenhofen a. d. Ilm, Germany
[3]Independent Scholar, Stuttgart, Germany
[4]Dipartimento di Agraria, Unversity of Sassari, Italy
[5]Faculty of Plastics Engineering and Surveying, University of Applied Sciences Würzburg-Schweinfurt, Germany

*Correspondence to*: Thomas Bueche (tbueche@web.de)

**Abstract.** Numerical modeling provides the opportunity to quantify the reaction of lakes on alterations in their environment, such as changes in climate or hydrological conditions. The one-dimensional hydrodynamic General Lake Model (GLM) is an open-source software and widely used within the limnological research community. Nevertheless, neither an interface to process the input data and run the model, nor tools for an automatic parameter calibration exist. Hence, we developed *glmGUI*, a Geographical User Interface (GUI) including a toolbox for an autocalibration, parameter sensitivity analysis, and several plot options. The tool is provided as a package for the freely available scientific code language R. The model parameters can be analyzed and calibrated for the simulation output variables water temperature and lake level.

The *glmGUI* package is tested for two sites (Lake Ammersee, Germany, and Lake Baratz, Italy) distinguishing in size, mixing regime, hydrology of the catchment area (i.e. the number of inflows and their runoff seasonality), and climatic conditions. A robust simulation of water temperature for both lakes (Ammersee: RMSE = 1.17 °C, Baratz: RMSE =1.30°C) is achieved by a quick automatic calibration. The quality of a water temperature simulation can be assessed immediately by a difference plot provided by *glmGUI*, which displays the distribution of the spatial (vertical) and temporal deviations. The calibration of the lake level simulations of Lake Ammersee for multiple hydrological inputs including also unknown inflows yielded a satisfactory model fit (RMSE = 0.20 m). This shows that GLM can be also used to estimate the water balance of lakes correctly. The tools provided by *glmGUI* enable a less time-consuming and simplified parameter optimization within the calibration process. Due to this, the free availability and the implementation in a GUI, the presented R package expands the application of GLM to a broader field of lake modeling research and even beyond limnological experts.

## 1 Introduction

As lakes response to changes in their environment they are often considered to be "sentinels of change" (Williamson et al., 2009, Hipsey et al., 2017). The investigation of alterations in the physical conditions of lakes, such as water temperature,



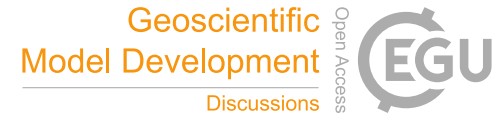

stratification, water balance, mixing behavior, or ice cover, has a key role in the understanding of the lake dynamics. Numerical modeling provides the opportunity of research beyond the analysis of observational monitoring data (Frassl et al., 2016) enabling simulations of periods without in-situ data and future conditions of lakes.

The development and application of community-based models is of increasing importance to find solutions for the future
challenges to simulate water body conditions under environmental alterations (e.g. climatic, land use, and agricultural policies, Bruce et al., 2018).The one-dimensional hydrodynamic General Lake Model (GLM) has been developed, under the leadership of members of the Global Lake Ecological Observatory Network (GLEON, gleon.org, Hanson et al., 2016) in response to the need of a robust model of lake dynamics (Hipsey et al., 2017), which is applicable for the vast diversity of lakes and reservoirs around the globe. GLM is able to simulate the thermal dynamics of lakes in their temporal and spatial
(vertical) characteristics. The model code is open-source and applied in numerous studies to a broad variety of different lakes and irrogations (e.g., Bueche et al., 2017; Bucak et al., 2018; Bruce et al., 2018; Fenocchi et al., 2018; Robertson et al., 2018; Ladwig et al., 2018; Mi et al., 2018; Fenocchi et al., 2017; Fenocchi et al., 2019). Carey and Gougis (2017) present the usage of GLM beyond the research application incorporated in a teaching tool for students.

Despite the high applicability of GLM a powerful toolbox for automatic calibration, validation and statistical sensitivity is
not yet existent. Therefore, we developed an R-based Geographical User Interface (GUI) implemented in the new R package *glmGUI* combining an easy handling of GLM simulations, a tool to automatize the calibration process, and visualization options for the input and output data. In general, the maxim of our project was inspired by four aspects:

- Provision of an open-source tool
- Provision of an usability friendly tool, which could be used by experts as well as less experienced modelers and
limnologists
- Using scripts to adapt the tool, with high acceptance and contribution in the scientific community
- Flexibility for the implementation of different calibration parameters and for the numerical and graphical interpretation of the output results

The R language (https://www.r-project.org) was chosen because it is open-source, flexible and an independent platform
(Snortheim et al., 2017). Moreover the lake modeler community uses already R based packages, i.e. *glmtools* for parameterization or plotting model output (https://github.com/USGS-R/glmtools) and *rLakeAnalyzer* for post-processing and evaluation of the model results (Winslow et al., 2016, https://github.com/GLEON/rLakeAnalyzer).

Changes in the water level of a lake can have strong influences on hydrodynamics, such as thermocline depth and stratification stability and duration, and can also affect lake water quality (Robertson et al., 2018). The appropriate water
level reproduction by lake models is essential for a robust simulation of spatial features of thermal dynamics in lakes, especially in shallow lakes with high variations in water stages. Furthermore, an accurate simulation of the lake level ensures the correct representation of hydrological interaction of the lake with its environment in the catchment area. Hence, in addition to the model output water temperature, the lake level is included to be calibrated in the provided automated calibration tool.



In this contribution we present the options included in *glmGUI* and show the application for two different sites, the pre-alpine deep Lake Ammersee, south Germany, and the shallow, Mediterranean Lake Baratz, Sardinia, Italy. The objectives of these two case studies are the calibration of the water temperature and lake level simulations of GLM using the automatic calibration tool.

**2 Lake model, software description, and toolbox options**

**2.1 The hydrodynamic lake model**

The GLM is a one-dimensional hydrodynamic model simulating the vertical profiles of temperature, salinity, and density at one spatial point in a lake over time (Frassl et al., 2016; Robertson et al., 2018). It applies the Lagrangian layer structure adapting the thickness and volume of layers with uniform properties from each simulation step (Bueche et al., 2017). The

underlying equations and hydrodynamics closures are documented in Hipsey et al. (2014) and Hipsey et al. (2017). The hydrodynamic model can also easily be coupled with the Aquatic Ecodynamics library (AED2) to simulate water quality simulations (Weber et al., 2017; Bruce et al., 2018).

GLM simulations are based on parameterizations of mixing processes, surface dynamics, and the effect of inflows and outflows. The model performance of simulating the lake thermal dynamics as well as the water balance can be improved by

a calibration of lake-specific parameters. The model documentation of Hipsey et al. (2017) includes also a description of the lake specific parameters and the default values for GLM simulations. The model requires meteorological and hydrological input data (see Table C1). Field data of water temperature should be available in and suitable temporal and spatial (several depths) resolution to enable a reliable calibration process.

**2.2 R package *glmGUI***

The R package *glmGUI* is a self-written extension in R interacting with the functionality of GLM. It provides two basic application functions - the logical elements of model-fit criteria calculations and graphical user interfaces for data visualization. The package requires a software version of R of $\geq 3.4$ and the version of *rLakeAnalyzer* $\geq 1.8.3$. All required R packages are automatically installed during the first installation of *glmGUI*.

**2.3 Graphical User Interface**

The Graphical User Interface is constructed as a window based design using functions of the R package *gWidgets* (Verzani, 2014), which provides several toolkits for producing user interfaces by creating e.g. labels, buttons, containers, or drop lists. The GUI is organized in a main menu with five sections. Submenus and result messages are opened in separate windows (Fig. 1).



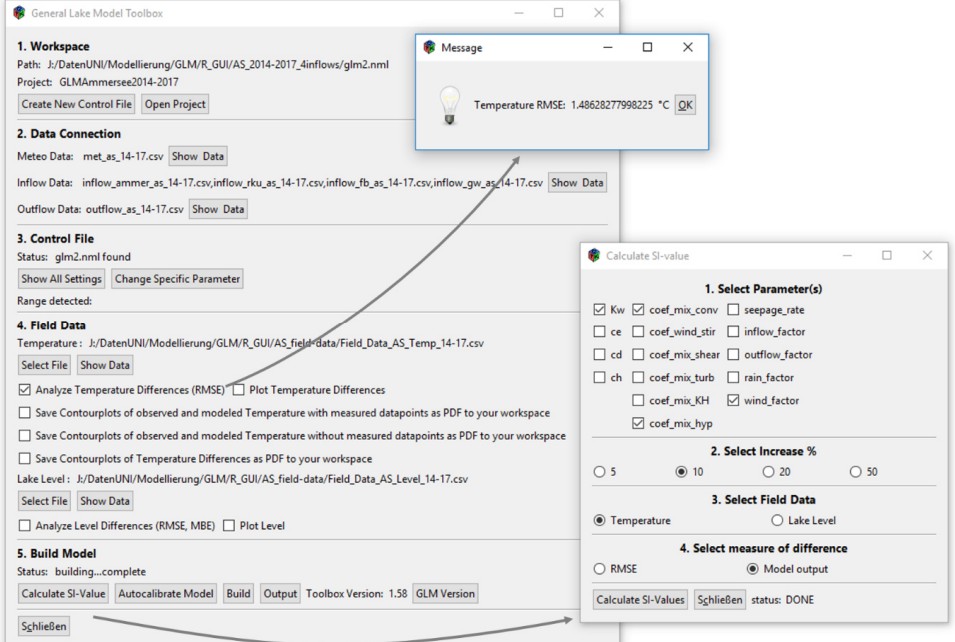

**Fig. 1: Window-based structure of glmGUI with examples of the submenu 'Calculate SI-value' and of the result message 'Temperature RMSE' created after a model run**

### 2.3.1 Data pre-processing and model efficiency

The GUI enables either the usage of an existing simulation control file of GLM (glm2.nml) from a selectable workspace (section 1, Fig. 1), or the creation of a new control file using the function nml_template_path() of the R package *GLMr* (https://github.com/GLEON/GLMr). The lake specific parameter and settings are according the deposited file of *GLMr* and is due to change with the current GLM version. The input data are automatically listed in section 2 (Fig. 1), if stored at the path as specified in the control file. All included time series of input data can be visualized and tested against missing values

(NA). The toolbox includes an option to fill the missing values applying the non-parametric Kalman-filter method (Grewal, 2011) using the R package *imputeTS* (Hyndman and Khandakar, 2007). If selected, an autofill option writes the interpolated values directly to the input file. All parameter settings defined within the control file can be shown and changed by the GUI (section 3, Fig. 1).

The model simulation can be run (section 5, Fig. 1) and several plot options can be selected to compare the model result with

observed field data (section 4, Fig. 1). As lake level variations will have a strong impact on the water temperature distribution within the lake, the validation of the lake level simulation results is provided within the GUI additionally to the





water temperature. The root mean square error (RMSE), which is often applied as model fit criteria in lake modeling studies (e.g. Bueche et al., 2017; Luo et al., 2018; Frassl et al., 2018), can be computed for both model output variables. Additionally, the mean bias error (MBE) is calculated for lake level simulations. Both model criteria are ascertained for all available observed data points and averaged subsequently

**2.3.2 Plots and output visualization**

Plot options are provided by *glmGUI* for the input time series in section 2 (Fig. 1) and for all output variables generated by GLM (csv and netCDF, section 5). This includes simple line plots (e.g. for lake level or evaporation) and contour plots for parameter varying in lake depth, such as water temperature or density. In addition, two types of contour plots of the vertical profile can be created. First is the visualization of observed and modeled water temperatures in one plot above each other

with the option of the measured data as point-overlay to mark where and when field data are available (areas in between are interpolated to draw the plots, Fig. 2). Second plot visualizes the temperature differences between the interpolated measured values and the modeled data. This plot type is a new feature enabling a quick overview on the spatial and temporal errors and deviations of the simulation (Fig. D1). The displayed deviations are fixed to 9 classes in the range of the errors between -5 °C to +5 °C and all values beyond these limits are shown in one color (≤ -5 in dark blue and ≥ in red) summarizing and

highlighting severe errors. The spatial reference in both plots is the lake surface, but lake level variations are described by changes in depth.

The generation of the contour plots is based on functions provided by *glmtools*. The default settings to scale the color bar legend for water temperature plots consider the range of temperatures and also erroneously the range of lake depth. This method is adopted in *glmGUI* and the temperature range is adjusted explicitly to the plotting method to provide well

differentiated color ranges in the legend.



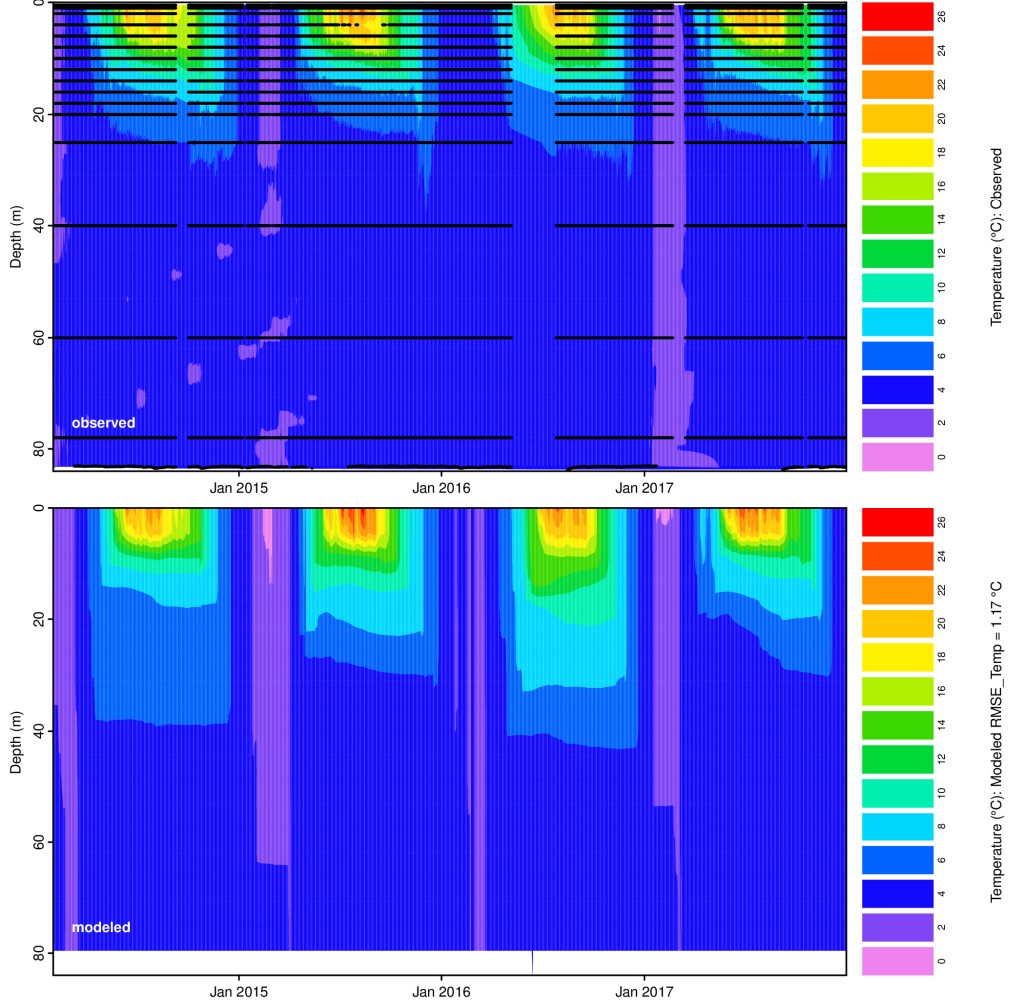

**Fig. 2: Example of a contour plot of observed (top) and modelled (below) water temperatures for Lake Ammersee. Black dots mark the availability (time/date and depth) of observed water temperatures**

### 2.3.4 Sensitivity analysis

5    To reduce the effort of the model calibration process only sensitive parameters should be included (Luo et al., 2018), which usually are identified applying a sensitivity analysis. It investigates how variations of input parameters or factors can be attributed to variations in the output of a numerical model (Pianosi et al., 2016). The widely used approach after Lenhart et



al. (2002) is implemented in the GUI and values of the Sensitivity Index (SI) are calculated for each selected parameter. It can be chosen out of four grades of relative changes of a parameter. The sensitivity of the simulation can be analyzed concerning the model output of water temperature or lake level. SI-values can be calculated based on either the respective model output or the RMSE, as also applied by Rigosi et al. (2011).

### 2.3.4 Autocalibration

Since the GLM uses empirical equations (Hipsey et al., 2017), model parameters can be adjusted during the calibration process to minimize the error between model output and observations (Luo et al., 2018). As an alternative to adjust the parameters manually in the glm2.nml-file, *glmGUI* provides an automatic calibration tool for preselected parameters of surface dynamics, mixing parameters, and hydrological and meteorological factors (Table C2). The user can choose out of these parameters those which are to be included in the calibration process and define a percentage range, by which the upper and lower limit of every parameter is changed from the value in the glm2.nml-file. The resolution of the increase/decrease of the parameters within the defined limits can be set as well. According to these settings, model runs of GLM are executed with all possible combinations of the selected parameters ("brute-force"). The overall RMSE of the lake level or water temperature is calculated and saved for every parameter combination to a csv file, so the "best fit" is indicated.

The automated calibration of the lake level includes also the optimization for the parameter *inflow_factor* for multiple lake inflows. This enables an approximation to the water balance and the reproduction of the lake level, if the contribution of inflows is unknown, which is often the case for groundwater inflows or smaller tributaries.

The runtime of the calibration algorithm (*t_cal*) increases exponentially with the number of parameters (*p*) to be calibrated (Eq.(**1**))

$$t_{cal} = (r + 1)^p * (t_{GLM} + t_{RMSE}) \qquad (1)$$

with *r* as resolution of variation within the set limits for each parameter and *t_GLM*, *t_RMSE* as runtimes of the lake model and the calculation of the output RMSE.

### 3 Case study Lake Baratz

### 3.1 Study site

Lake Baratz is located in the northwest of Sardinia (Fig. 3), Italy, and is the only natural lake of the island. The elevation of its bottom is 18.6 m a.s.l. and the lake level suffered significant changes in the last two decades with a maximum lake depth of 11 m and a minimum of 3 m (Giadrossich et al., 2015; Niedda et al., 2014). The overflow spillway of the lake is at 32.5 m a.s.l. (Niedda et al., 2014). At this maximum level the lake has a surface of about 0.6 km² and volume of $5.1 \times 10^6$ m³





(Giadrossich et al., 2015). As lake-overflow events to the sea were extremely rare in the past century the catchment area can be considered as a closed-basin (Niedda and Pirastru, 2013). The lake watershed is about 12 km² with a maximum elevation of 410 m (Pirastru and Niedda, 2013). The only significant tributary is inflowing the lake in the northeast and drains a sub-catchment area of 8.1 km². Due to a very dry summer season the water inflow starts usually in December and ends in May (Giadrossich et al., 2015). The sub-catchment of the Grifone gauging station has an area of 7.4 km².

The lake can be classified as eutrophic and the water is brackish. Thermal stratification usually establishes in February or early March and lasts to early autumn (Giadrossich et al., 2015). Lake mixing occurs all throughout during winter and thus, it can be classified as a warm monomictic lake.

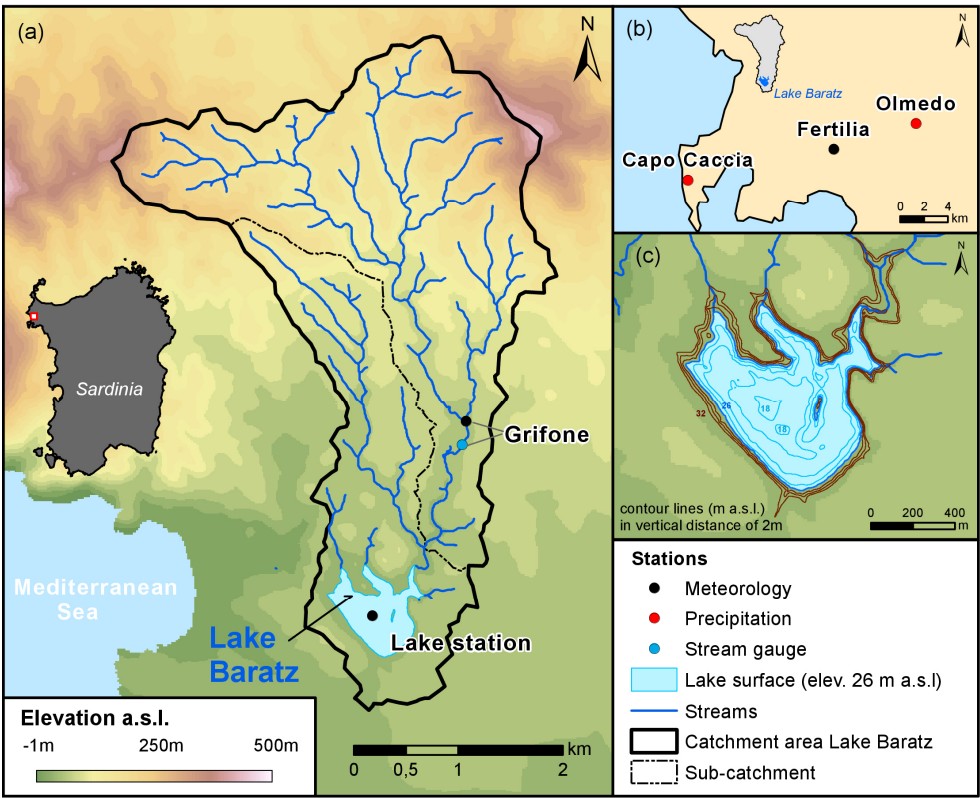

**Fig. 3:** a) Lake Baratz and its hydrological catchment area and observation station, b) Observation stations outside of the lake watershed (grey area), c) Contour lines of the lake basin. The blue area indicates the lake surface at a lake level of 8 m (equal to an elevation of 26 m a.s.l) observed in November 2012. Brown lines display dry terrain and cyan isobaths at this lake level.





### 3.2 Sensitivity analysis and calibration

The simulation period for Lake Baratz is determined to be 13.07.2011 to 31.12.2016. Although input and field data are available after 2016, this period is chosen considering a distinctive reduced transparency of water for 2017, indicated by Secchi-disk observations, and the computed average $K_w$ value of 0.57 m$^{-1}$ (period 2011 – 2016) would not be representative

for this year. Salinity values for the initial profiles defined for the simulation start are derived from conductivity data surveyed on the 21$^{st}$ June 2011. A further detailed description on the applied meteorological and hydrological model input data, the field data, and the data processing can be found in the Appendix A.

As preliminary estimations of the water balance considering the seasonality of the inflow and subsurface outflow already exist (see Appendix A3.3), which are also applied in this study, the calibration process of the lake level was accomplished

without hydrological parameters. Thus, the sensitivity analysis was performed considering only the parameters of surface dynamics and the *wind_factor* and the options of 10 % increase and the RMSE as measure of difference were selected.

After Lenhart et al. (2002) only one of four parameters (*cd*) is found to have a negligible sensitivity indicated by a SI-value of below 0.01. The analysis revels a high sensitivity for the parameters *ch* (SI = 0.234, see also Fig. 4) and *ce* (SI = 0.334) and a medium for *wind_factor* (SI = 0.089).

The sensitivity analysis regarding water temperature was conducted for parameters of lake mixing, surface dynamics, and the wind factor with the same options as selected for the lake level. Negligible SI-values of 0.005 and below are found for all considered parameters except for *ce* (SI = 0.232) and *wind_factor* (0.290) with a high sensitivity, and *ch* (0.050) with a sensitivity at the threshold between medium and small (Fig. 4). According to both sensitivity analyses the model is calibrated first for the lake level considering the parameters with medium and high sensitivity. As the model is found to be sensitive in

lake level and water temperature simulations for changes in the same parameters, the calibration for water temperature simulations was performed only considering *ce* and *wind_factor* to prevent a decline of the lake level reproduction by the water temperature calibration. In addition, small parameter value ranges of 10 % are applied.





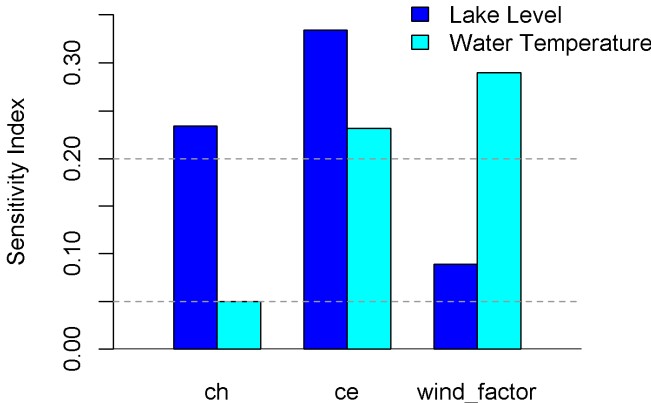

**Fig. 4: Sensitivity Index after Lenhart et al. (2002) of GLM simulations of Lake Baratz regarding the lake level (blue) and water temperature (cyan). The lower horizontal line (SI = 0.05) marks the threshold between small and medium sensitivity, the upper line (SI = 0.20) between medium and high sensitivity.3.3 Simulation results**

The calibration processes for lake level simulations of the lake model reveals the best fit applying an unchanged parameter value for *ch* and an adjustment of *ce* to 0.001748 and of *wind_factor* to 1.56. The simulated lake level of Lake Baratz shows very good model fit criteria of an average RMSE = 0.11 m and an average MBE = 0.05 m. The seasonal pattern of the lake level, characterized by a strong increase during winter and a drop during summer, is traced well. Also the general decrease of the water stage by about 3 m during the simulation period is simulated correctly, which proofs the capability of the model to reproduce the water balance of the lake and its catchment area.




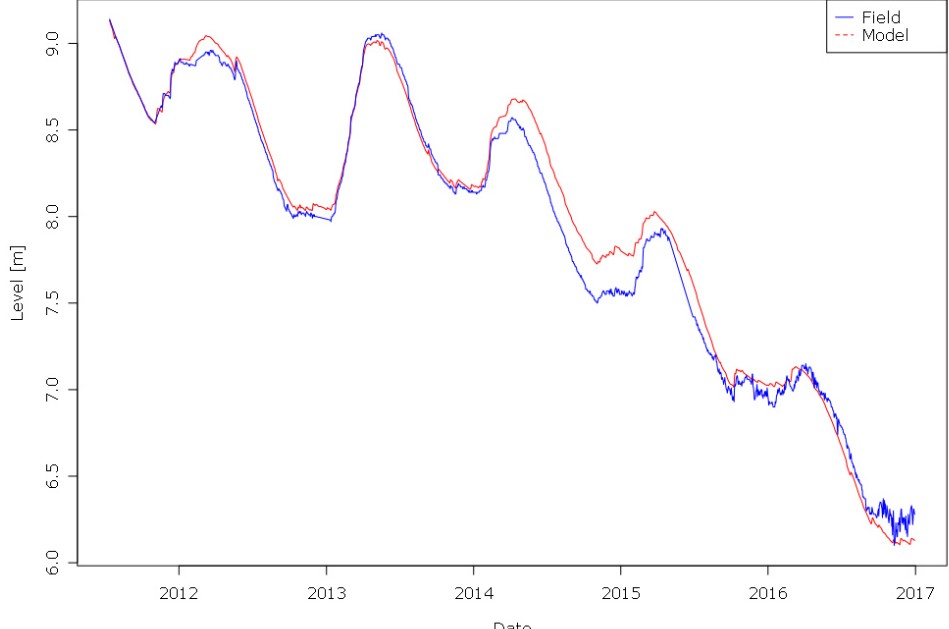

**Fig. 5: Observed (blue) and simulated (red) lake level of Lake Baratz.**

The modeled water temperatures simulated with the adapted lake parameters found by the (auto-)calibration yield an overall average RMSE of 1.30 °C. The model reproduces stratified and isothermal conditions (Fig. 6). A detailed overview on the
5  temporal and spatial differences between the observed and simulated water temperatures is given in Fig. D1. The simulation can be assessed as a robust fit (Bruce et al., 2018) especially when considering the high degree of thermal dynamics due to the shallow depth of the lake and an additional enhancement by highly variable lake level (Fig. 6). The results approve also GLM to be applicable for shallow lakes in warm climates with high variety in lake level.





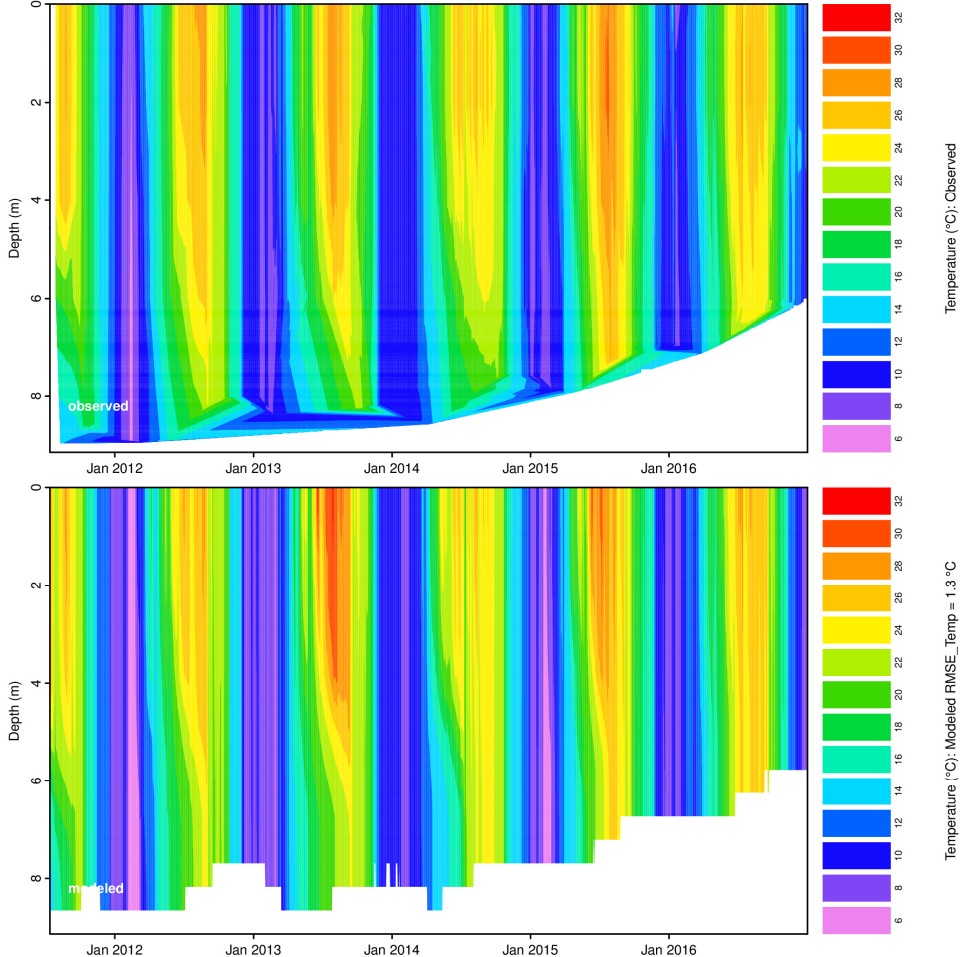

**Fig. 6: Observed (top) and simulated (bottom) water temperatures of Lake Baratz**



## 4 Case study Lake Ammersee

### 4.1 Study site

The pre-alpine Lake Ammersee (Fig. 7) has a maximum depth of 83.7 m, a surface area of 46.6 km² and a volume is about $1.8 \times 10^9$ km³ (Bueche and Vetter, 2014b). The mixing regime can be classified as dimictic, but also monomictic seasons

5   occur (Bueche, 2016). The trophic status is currently mesotrophic (Vetter and Sousa, 2012).

The lake outflow in the north (Stegen gauge station) has a catchment area of about 994 km². The main tributary is River Ammer, which contributes approximately 80% of the total annual discharge to the lake (Bueche and Vetter, 2015). Several other streams and creeks inflow into the lake, but only River Rott and Fischbach have a share of greater than 5 % of the total lake catchment area size (see Table B1). Additionally, groundwater is assumed to inflow the lake, which has not been

10   quantified yet (Bueche and Vetter, 2014a). The mean lake level is 532.9 m a.s.l. and usually varies about 1 m (Bay. LfU, 2018).





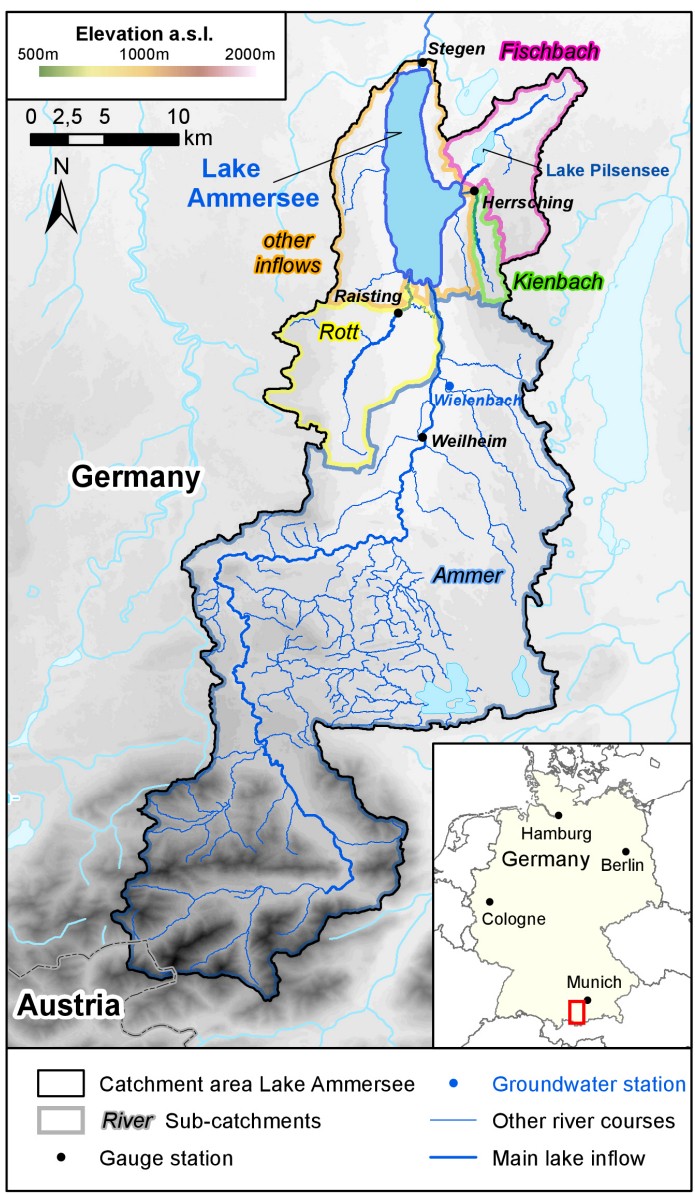





**Fig. 7: Catchment area (with sub-catchments) of Lake Ammersee. Dots mark the locations of hydrological observation stations (Source DEM: Elevation data from ASTER GDEM, a product of METI and NASA, Source geo-data: Geobasisdaten ©Bayerische Vermessungsverwaltung, www.geodaten.bayern.de)**

**4.2 Sensitivity analysis and calibration**

The simulation period for Lake Ammersee is chosen to be 30.01.2014 to 31.12.2017 starting when reliable field data of the lake station are available consistently (see Appendix B). The initial profile of water temperatures are taken from the observations of that date. No water quality data were available for the simulation period and the salinity values are derived from conductivity measurements of January 2004, when similar thermal conditions of a slight inverse stratification were present and equivalent salinity conditions can be assumed. As the trophic status has not changed since 2004 the average

value $K_W$ of 0.35 m$^{-1}$ calculated for the period 2004 to 2008 from secchi-disk observations (data surveyed and provided by Bavarian Environment Agency) is used for the GLM simulations in this study.

The calibration of the lake level considers the factors of discharge of the four defined inflows of River Ammer, River Fischbach, groundwater inflow, and the sum of River Rott, River Kienbach, and all other unknown inflows. Thus, the adjustment of the representative *inflow_factor* includes the required correction of the available discharge data considering

the observations are taken not at the stream inlet to the lake but at an upstream location. This is especially relevant for River Ammer with its gauge at Weilheim (Fig. 7). Meteorological input data are taken by a raft station at the lake center except, precipitation and cloud cover data (see Appendix B for a detailed description about the sources of the used meteorological and hydrological input data and the processing of the data).

The sensitivity analysis regarding the water temperature (increase: 10%, measure of difference: RMSE) reveals seven

parameters with an SI > 0.05 indicating a medium sensitivity (after Lenhart et al., 2002, Table 1). In order to reduce the calculation time of the autocalibration runs, four parameters of high sensitivity with a SI above 0.2 are chosen for this process.

**Table 1: Values of SI for Lake Ammersee water temperature simulations**

| Parameter | SI |
|---|---|
| ch | 0.534 |
| coef_mix_shear | 0.484 |
| coef_mix_turb | 0.410 |
| wind_factor | 0.210 |
| cd | 0.126 |
| coef_mix_KH | 0.097 |
| coef_wind_stir | 0.060 |
| ce | 0.034 |
| coef_mix_conv | 0.015 |
| coef_mix_hyp | 0.008 |





### 4.3 Simulation results

The calibration of the lake level simulation yields its best fit for the combination of the inflow factors for the defined tributaries River Ammer of 1.10, River Fischbach of 0.72, groundwater of 1.07, and Rivers Rott, Kienbach and all other smaller and unknown inflows of 1.01.The overall RMSE reduced significantly from 1.10 m to 0.20 m, and the MBE from -

5    1.00 m to 0.09 m, and the achieved model fit can be assessed as very satisfactory. The simulation shows periods of general deviations of over 0.20 m up to 0.55 m for some months, but reproduces well the short-term fluctuations (Fig. 8). The remaining errors and differences can be ascribed to the uncertainties and lack of data for some inflows as assumptions and estimations for the unknown surface input and the groundwater inflow had to be made. More detailed hydrological data might explain remarkable dates, like during summer, when the trend of simulated and observed lake level changes abruptly.

10   However, it was possible to improve the lake level simulation distinctively considering only the parameter *inflow_factor* for multiple inflows within the calibration process. Due to the very detailed pre-processing the used input and field data are very reliable and can be excluded as source of high model errors.

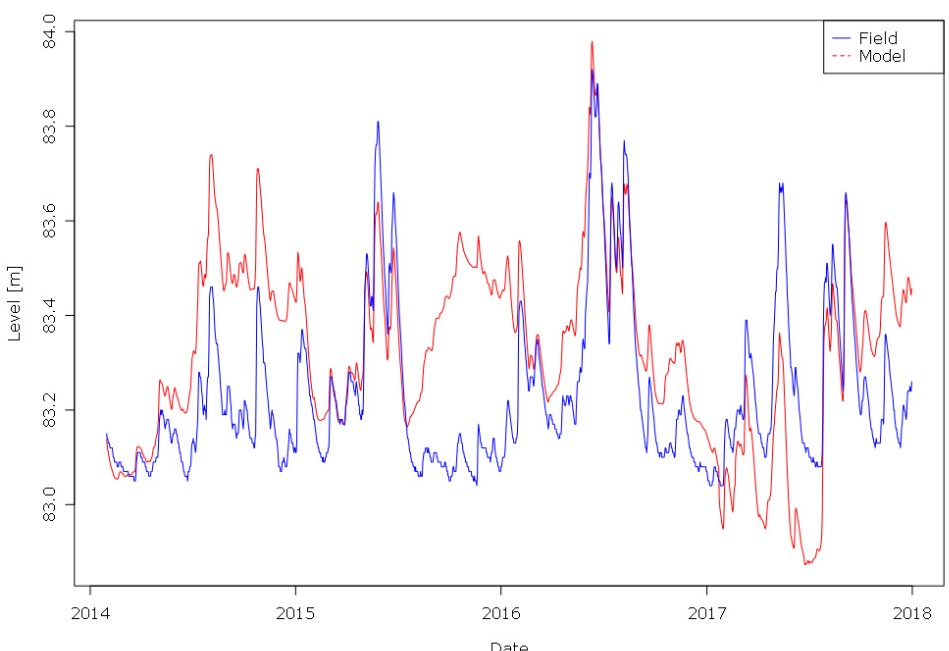

**Fig. 8: Observed (blue) and simulated (red) lake level of Lake Ammersee**



Simulated water temperatures reveal an overall RMSE of 1.172 °C. The highest deviations occur within the metalimnion in a depth between 7 – 18 m (Bueche and Vetter, 2014b) during summer stratification, when the model underestimates the temperature (Fig. 9). This is accompanied by too warm layers in the upper epilimnion in a depth around 20 m. However, the general seasonal pattern is reproduced satisfactory and differences during winter are small.

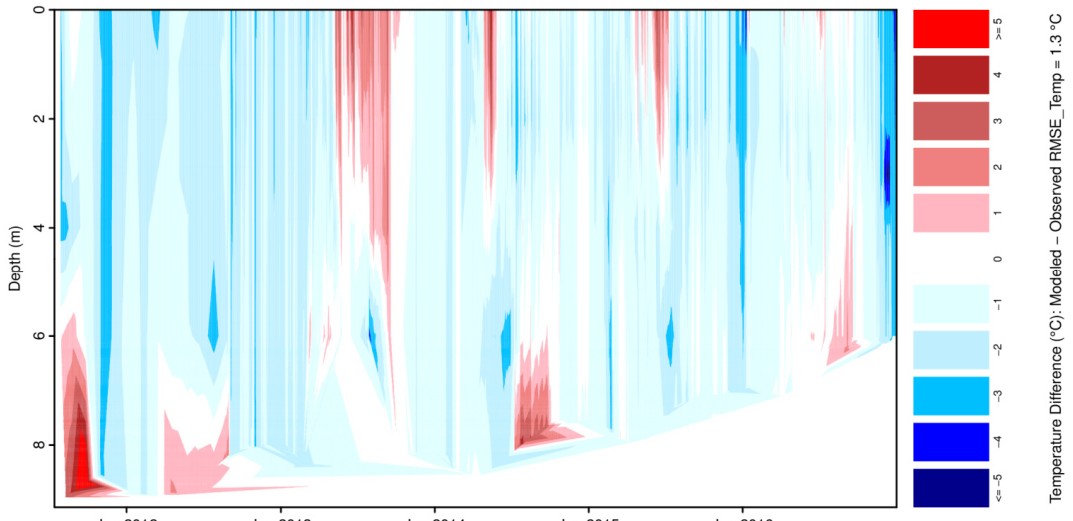

**Fig. 9: Contour plot of differences between simulated and observed water temperatures of Lake Ammersee**

## 5 Discussion

The implemented option of an autocalibration in the toolbox enables a less time-consuming and more efficient parameter optimization than a conventional manual calibration procedure (Luo et al., 2018). The utilization of such techniques of automatization is advised for lake modeling studies (Ladwig et al., 2018). Hence, the provided tool can be seen as the centerpeace of the developed GLM Toolbox, which is complemented by the plotting option of differences between observed and simulated water temperatures. This model output visualization enables an immediate overview on the simulated deviations and their spatial distribution without any further post-processing of the model results. Although no detailed quantification of the error is possible, this illustration allows a very quick qualitative comparison of different simulation settings. Such visualization option for GLM output has not been provided for an open-source software before.

The easy handling and free availability of the GUI expands the reach of potential user of the GLM beyond the limnological research specialists. The used scripting language R is already widespread in limnological (e.g. Winslow et al., 2016), hydrological and environmental research (e.g Pilz et al., 2017; Gampe et al., 2016), as well as for the automation of water





management processes (e.g. Erban et al., 2018). Providing the R code as development version (http://doi.org/10.5281/zenodo.2025865) in addition to the R package, the GUI and its tools can be easily customized by users for other specific demands and then again be shared with the public.

Due to the small number of considered parameters in the calibration process of the presented case studies, the efficiency was
successfully tested for a realistic effort of time. The visibility of the simulation error provided by the created difference contour plots gives the opportunity to combine the automated calibration easily with expert knowledge. However, the time consumption of an autocalibration run of several days and more for combinations of larger number of parameters and higher intervals might pose a restriction of the calibration efficiency of a solely automatized calibration and can be improved in upcoming versions of the toolbox. The expectable improvement of computer hardware performance in the future could also
compensate this problem.

Further limitations of the toolbox/GUI have to be mentioned and more coding activities should be addressed to minimize the following issues. 1) The contour plots of water temperatures indicate the lake level on the bottom either interpolated or in very coarse lake level fluctuation, which does not allow a sufficient derivation of this model output by these plots. For this purpose separated plots of the lake level simulations can be created. 2) The calibration algorithm of the toolbox creates
parameter values with a high decimal precision due to the approach of percental alteration. This might pretend a false sensitivity of the model and a too detailed accuracy of the calibration for the respective parameter. 3) The method of the sensitivity analysis will yield a SI = 0 for any included parameter with the initial value is 0. This applies for example to *seepage_rate* as its default value is 0. 4) The applied model version of GLM is dependent of the maintenance of the R package *glmtools*.

The presented simulations of lake water temperatures have an average overall RMSE of 1.30 °C and 1.17 °C. This is in the range of values obtained by other lake modeling studies applying GLM or other lake 1-D models (Bueche et al., 2017; Ladwig et al., 2018; Robertson et al., 2018; Frassl et al., 2018). This simulation quality was achieved even by only a few autocalibration runs showing the expedience of the tool.

In addition to the visualization options and the calibration tool for water temperature simulations, *glmGUI* enables also a
calibration of the lake level to achieve a correct reproduction of the water stage. This is especially important for smaller lakes, for which an incorrect simulation of the lake level can have a strong impact on the water temperature reproduction. Furthermore, as the lake level is the result of the lake-catchment water balance (Vanderkelen et al., 2018), the applicability of GLM is enhanced also to hydrological analysis and water balance investigations by this feature. In this study, a RMSE for the lake level simulations of 0.11 m (Lake Baratz) and 0.20 m (Lake Ammersee) is achieved, which is in the range of the
GLM performance shown by Weber et al. (2017), and attests a good agreement of the modeled water level to the observed values (Hostetler, 1990).

The GUI is also able to execute simulation runs of GLM coupled with water quality models of ecological lake models (e.g. Aquatic Ecodynamics Modelling Library, AED). Although water quality settings cannot be changed so far using the toolbox, any model generated output can be plotted.





## 5 Conclusions

The presented R package *glmGUI* wraps up simulation and processing tools for the GLM. This includes a tool to autocalibrate the model and options of a parameter sensitivity analysis. Both tools can be used to examine the two simulation output variables of water temperature and lake level. Furthermore, *glmGUI* implements several visualizations options for the

meteorological and hydrological input data and the model output. After the deployment of other R packages to execute GLM and for model output post-processing and statistical analysis (*rLakeAnalyzer*, *rGLM*, *glmtools*) *glmGUI* fills the gap of lacking tools to simplify and accelerate the calibration process and to extend visualization options.

The tools are tested for two different lakes (deep, dimictic, perennial inflow, and shallow, monomictic, seasonal inflow) located in varying climate zones. Good model results were achieved after a low expenditure of calibration effort. In contrast

to many other studies an exhaustive description of the simulation input data and field data (data for Lake Ammersee also provided as example data) is given within this paper. This includes a data quality assessment, detailed descriptions of the required pre-processing steps, and the sources of the implemented data.

The development of *glmGUI* for the free and open-source programming language R, available on all common platforms (Windows, OS X, Linux), makes it accessible for anyone to use, which contributes to scientific transparency (Winslow et al.,

2018). High interoperability and flexibility is given regarding other study cases, other time steps, or different scenario aims. This includes the coupling of GLM with ecological lake models (e.g. Snortheim et al., 2017; Robertson et al., 2018; Weber et al., 2017; Fenocchi et al., 2019), as the toolbox already works for this purpose and can be the basis for a development of a coupling interface. An increasing number of lake modeling studies are investigating the impact of global change on lakes using as meteorological input data regional climate model output or future scenarios (Fenocchi et al., 2018; Weinberger and

Vetter, 2014; Bueche and Vetter, 2015; Ladwig et al., 2018; Pietikäinen et al., 2018; Piccolroaz and Toffolon, 2018). For analyses in field of research the presented R package *glmGUI* will be a powerful tool, especially the provided difference contour plots.

*Code and data availability:* The package and R-code are available at http://doi.org/10.5281/zenodo.2025865. Example data

for Lake Ammersee are attached to this paper as a supplementary data. Sources are described in Appendix B.

*Author contributions:* TB and MV contributed the idea and concept for *glmGUI*. MW and BP developed the *glmGUI* package and code. The study sites were suggested by TB and FG. FG and MP operated the raft station and data collection at Lake Baratz. Data pre-processing was conducted by TB with support of MP, FG and BP. Simulations and analyses have

been conducted by TB with support of FG, MP and MV. The manuscript was prepared by TB with contributions of all co-authors.

*Data availability:*





**Acknowledgements**

We are grateful to the University of Sassari for founding one month of Professorship of TB in Sassari. We thank for the financial support of the study by the teaching and research program Lehre@LMU sponsored by the German Federal Ministry of Education and Research (BMBF) (grant no 01PL17016) and by the Mentoring Programme of LMU Munich. We also thank Bachisio Padedda for providing ecological data observations of Lake Baratz and Helge Olberding for assisting work. Finally, we pay a silent tribute to Marcello Niedda, who initiated and expedited the hydrological investigations at Lake Baratz.

**Appendix A: Input data for Lake Baratz**

**A1 Observation stations**

Meteorological data are taken from several stations in the environment of the lake including a observation station rafting on the lake surface center. Hydrological data (discharge and water temperature) were surveyed at a stream gauge at the Grifone site, where the observation setup was demolished on 31.05.2017. Fig. 3 gives an overview of the locations of the stations. The lake station was not in operation from 24.09.2013 to 25.04.2014 (Giadrossich et al., 2015).

**A2 Meteorological model input data**

**A2.1 Air temperature**

Input data for air temperature are taken from the respective station in the following order:

- Lake station ($R^2 > 0.93$, reference period: 08.07.2011 – 23.09.2013, Giadrossich et al., 2015)
- Calculated from Grifone station by linear regression to the lake station ($R^2 = 0.97$, reference period: 25.04.2014 – 31.05.2017, Fig. A1a)
- Calculated from Fertilia station by linear regression to the lake station ($R^2 = 0.99$, reference period: 15.01.2015 – 12.02.2018, Fig. A1b)





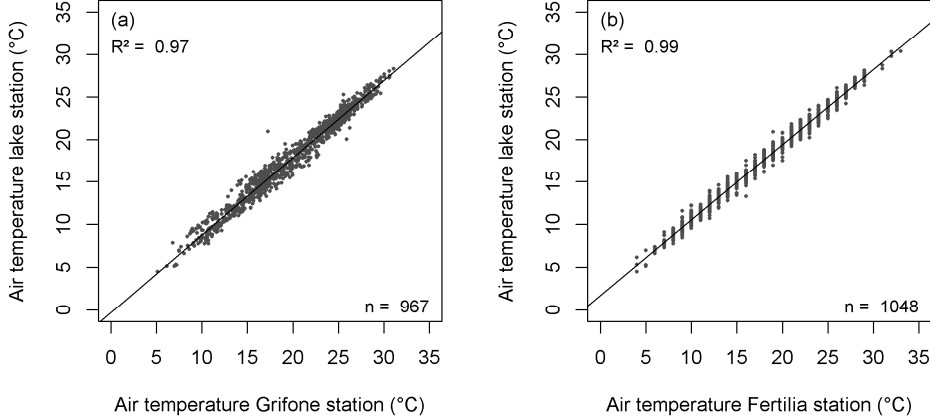

**Fig. A1: Linear correlation of air temperature for the lake station and a) Grifone station and b) Fertilia station.**

**A2.2 Wind speed**

Wind speed data at the lake station have several gaps of observations and the measurements exhibit a significant bias

5   between the periods before and after 21.06.2016 (Fig. A2), when a new sensor was installed after an outage. The bias is detected by comparing with data observed at Fertilia station. For the period before this date the average wind speeds measured at the lake station were by 1.19 ms[-1] lower than measured at Fertilia station. After the 21.06.2016 the difference was only 0.08 ms[-1].



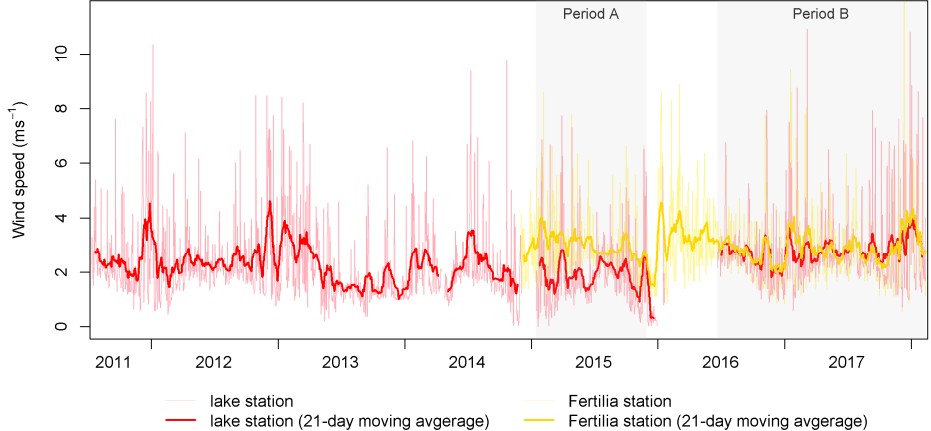

**Fig. A2: Time series of daily wind speeds at lake and Fertilia station. Periods A and B are used for mean value comparison (see Table A1: Averages of wind speed at lake and Fertilia station**

Table A1: Averages of wind speed at lake and Fertilia station

| Period | lake station | Fertilia station |
|---|---|---|
| 15.01.2015 - 30.12.2015 (period A, Fig. A2) | 1.82 | 3.01 |
| 21.06.2016 - 12.02.2018 (period B, Fig. A2) | 2.71 | 2.79 |

The final time series of wind speed input data (daily values) were prepared by first filling gaps of observations by linear correlation (Table A2), and secondly by adjustment of the values after the 21.06.2016 by multiplying with the factor of 0.67, which is equal to the quotient of the average data for periods before and after at the raft station (see Table A1).

**Table A2: Gaps of observations for wind speed at the lake station, station from which data were used, and coefficient of**
10 **correlation**

| Period of data gap | Data source | Reference period | R² | Comment |
|---|---|---|---|---|
| 24.09.2013 – 24.04.2014 | Grifone | 08.07.2011 – 23.09.2013 | 0.72 | (Giadrossich et al., 2015) |
| 04.12.2014 – 14.01.2015 | Fertilia | 21.06.2016 – 12.02.2018 | 0.53 | see Fig. A3a |
| 31.05.2015 – 20.06.2016 | Fertilia | 15.01.2015 – 30.12.2015 | 0.41 | see Fig. A3b |

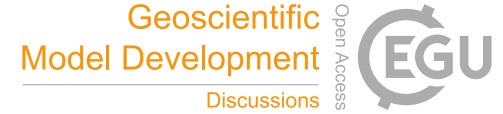



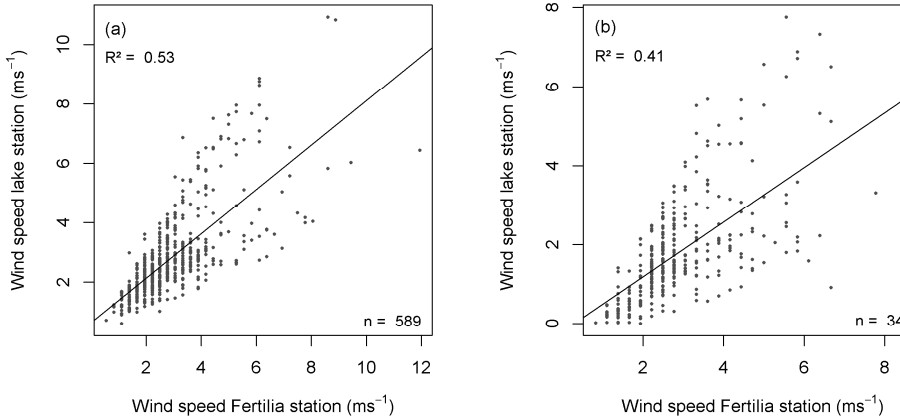

**Fig. A3: Linear correlation of wind speed for the lake and Fertilia station, a) period 21.06.2016 to 11.02.2018, b) 15.01.2015 to 30.12.2015**

5 **A2.3 Relative humidity**

Missing data for the lake station were filled by linear correlation. Table A3 gives an overview on the filled gaps and the used data source. The maximum observed value for relative humidity at the lake station is 96.89 %. Values computed form the other stations may exceed this value and were then set to be 96.7 %.

**Table A3: Gaps of observations for relative humidity at the lake station, station from which data were used, and coefficient of**
10 **correlation**

| Period of data gap | Data source | Reference period | R² | Comment |
|---|---|---|---|---|
| 24.09.2013 – 24.04.2014 | Grifone | 08.07.2011 – 23.09.2013 | 0.82 | (Giadrossich et al., 2015) |
| 04.12.2014 – 14.01.2015 | Grifone | 24.04.2014 – 03.12.2014 | 0.80 | see Fig. A4a |
| 21.01.2016 – 22.03.2016 | Grifone | 08.10.2015 – 23.11.2016 | 0.74 | see Fig. A4b |
| 24.11.2016 – 22.03.2017 | Fertilia | 24.03.2017 – 20.11.2017 | 0.89 | see Fig. A4c |



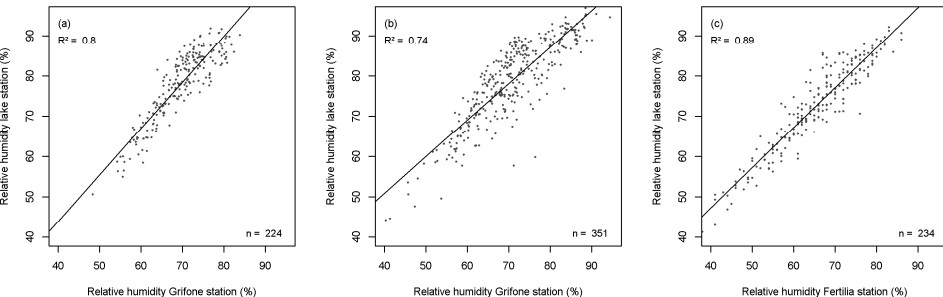

**Fig. A4: Linear correlation of relative humidity for the lake station and Grifone station (a) period 24.04.2014 – 03.12.2014 and b) period 08.10.2015 – 23.11.2016) and c) Fertilia station.**

### A2.4 Rainfall

5   No precipitation data are observed at the lake station. Measurements exist at Grifone site until 31.05.2017. Table A4 shows the gaps in the rainfall time series at Grifone station and the source for filling.

**Table A4: Gaps of observations for rainfall at Grifone and information on filling method**

| Period of data gap | Number of days | Data source (rainfall data) | Filling method/comment |
|---|---|---|---|
| 18.08.2013 – 23.08.2013 | 6 | - | assumed to be 0 |
| 14.09.2015 – 07.10.2015 | 24 | - | Estimated from soil moisture data at Grifone station |
| 28.11.2016 – 10.12.2016 | 13 | Olmedo & Capo Caccia | Average of the two stations |
| 15.12.2016 – 20.12.2016 | 6 | Olmedo & Capo Caccia | Average of the two stations |
| 12.01.2017 – 09.03.2017 | 57 | Olmedo & Capo Caccia | Average of the two stations |
| 10.03.2017 – 22.03.2017 | 13 | *Not filled* | |
| 01.06.2017 – 31.08.2017 | 92 | - | Assumed to be 0 |
| 01.09.2017 – 12.02.2018 | 193 | Capo Caccia | |

The mean annual precipitation at Grifone is about 600 mm (Pirastru and Niedda, 2013) and Sardinia is characterized by a

10   rainy winter season and dry summer months (Niedda et al., 2014; Chessa et al., 1999). Hence, periods in summer with no rainfall observations are assumed to be 0.

### A2.5 Shortwave and longwave radiation

Radiation measurements representing the net radiation are available at lake station and Grifone station. All data gaps at the lake station are filled by linear correlation from Grifone ($R^2 = 0.93$, Fig. A5). Three short gaps with maximum of eight days



are filled by linear interpolation. The required (incoming) shortwave $R_{Sin}$ (Wm$^{-2}$) and longwave (net) radiation $R_{Lnet}$ (Wm$^{-2}$) are calculated from the net radiation values using the energy balance:

$$R_n = (1 - \alpha) * R_{sin} + R_{Lnet} \tag{2}$$

where $R_n$ is the net radiation (Wm$^{-2}$), $\alpha$ is the albedo of the water surface (assumed to be 0.2, Hipsey et al., 2014). Net longwave is described as (Hipsey et al., 2017):

$$R_{Lnet} = R_{Lin} - R_{Lout} \tag{3}$$

5   where $R_{Lout}$ (Wm$^{-2}$) is the outgoing longwave radiation and $R_{Lin}$ (Wm$^{-2}$) is the incoming radiation. Based on the Stefan-Boltzmann law the incoming and outgoing longwave can be calculated from the emissivity and temperature of water and air, respectively (An et al., 2017):

$$R_{Lout} = \varepsilon_w * \sigma * (T_s + 273.1)^4 \tag{4}$$

$$R_{Lin} = \varepsilon_a * \sigma * (T_a + 273.1)^4 \tag{5}$$

where $\varepsilon_W$ the emissivity of the water surface, assumed to be 0.985 (Hipsey et al., 2017), $\varepsilon_a$ is the air emmisivity, $\sigma$ is the Stefan-Boltzman constant, $T_S$ is the surface water temperature (°C), and $T_a$ is the air temperature (°C). For $T_s$ can be used

10   measurements from the lake surface. For days with no observations $T_s$ is calculated from $T_a$ by polynomial regression (R² = 0.88, Fig. A6):

$$y = -6.154 * 10^{-7} * x^6 + 7.048 * 10^{-5} * x^5 - 3.139 * 10^{-3} * x^4 + 6.572 * 10^{-2} * x^3 - 0.620 \\ * x^2 + 2.923 * x + 3.536 \tag{6}$$

Air emissivity is calculated using the expression proposed by An et al., (2017, adopted from Idso, 1981):

$$\varepsilon_a = 0.7 + 5.95 * 10^{-4} * e_a * e^{(1500*(T_a-273.1)^{-1})} \tag{7}$$

$$e_a = \frac{RH}{100} * e_s \tag{8}$$

$$e_s = 0.6107 * e^{17.269*T_a*(T_a+273.1)^{-1}} \tag{9}$$

where $e_a$ is the vapor pressure (kPa), RH is the relative humidity (%) of air, and $e_s$ is the saturated vapor pressure (kPa) at $T_a$.





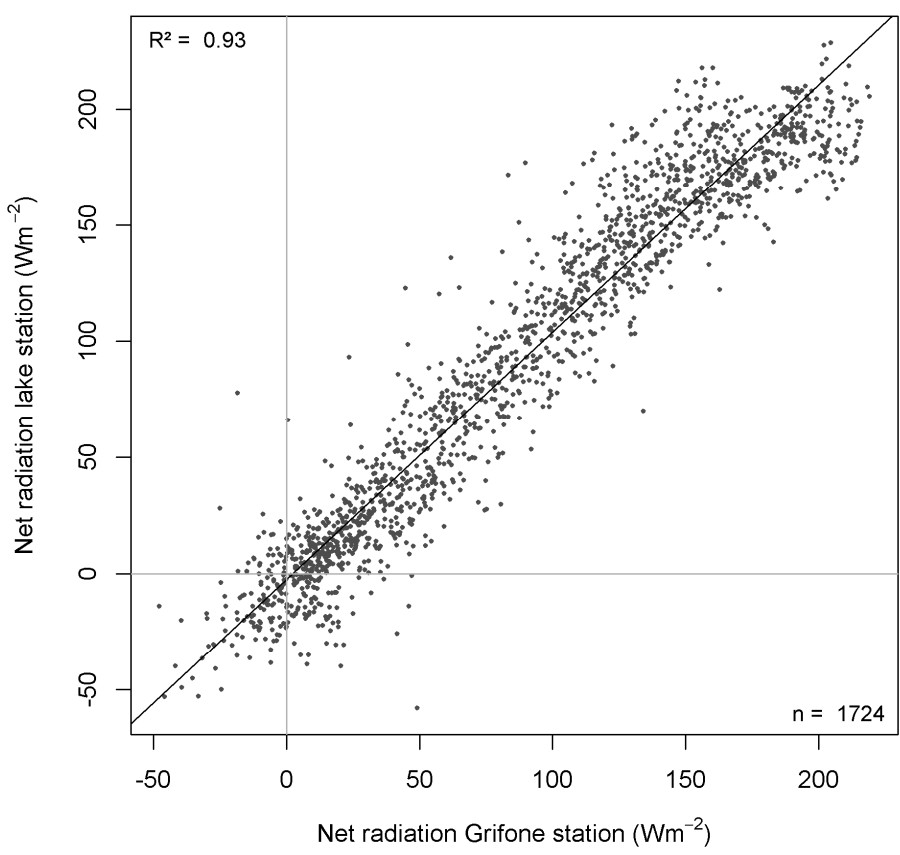

**Fig. A5: Linear correlation of net radiation for the lake station and Grifone station (period: 13.07.2011 – 30.05.2017)**



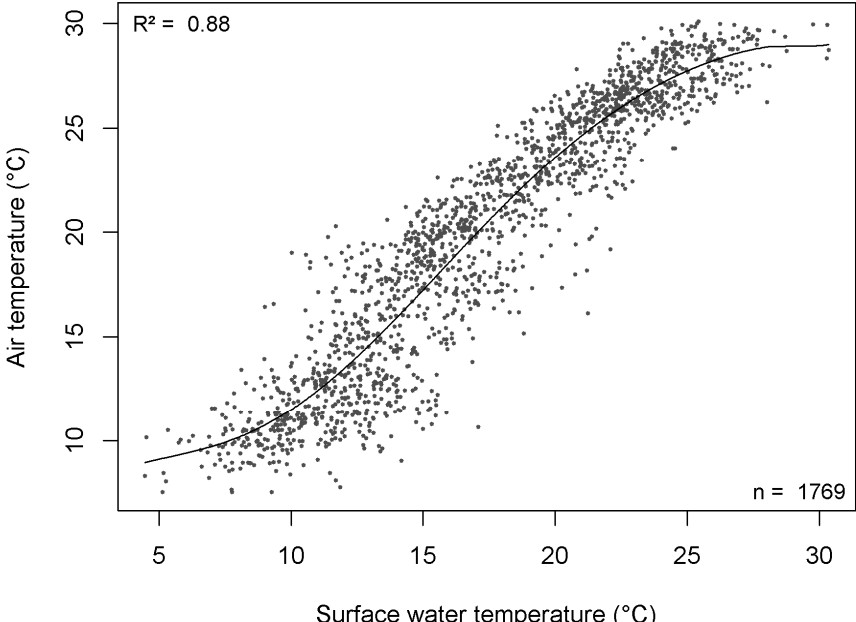

**Fig. A6: Polynomial correlation for the surface water and air temperature at the lake station (period: 13.07.2011 – 12.08.2018)**

The monthly means of the calculated shortwave radiation exhibit only slight deviations to the values presented by Lavangini et al. (1990, Table A5)

5   **Table A5: Monthly mean of shortwave radiation**

| month | Shortwave radiation (Wm$^{-2}$) | |
|---|---|---|
| | Lavagnini et al. (1990) | calculated |
| Jan | 78.1 | 91.4 |
| Feb | 110.0 | 120.0 |
| Mar | 167.8 | 181.3 |
| Apr | 225.7 | 242.5 |
| May | 289.4 | 293.4 |
| Jun | 312.5 | 318.9 |
| Jul | 312.5 | 309.2 |
| Aug | 277.8 | 280.4 |

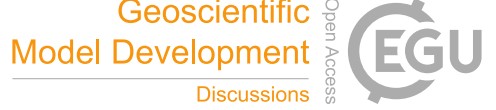



| | | |
|------|-------|-------|
| Sep | 223.4 | 211.3 |
| Oct | 148.1 | 148.3 |
| Nov | 81.0 | 95.3 |
| Dec | 67.1 | 83.0 |

### A3 Hydrological model input data

#### A3.1 Inflow discharge

Observed discharge data are available for Grifone site. At the cross-section the sub-catchment area of this gauge is 7.4 km²,

which represents approximately 62% of the lake watershed. The inflow to the lake is simulated applying the hydrological model developed by Niedda et al. (2014) representing the entire lake catchment. The daily values of the inflow discharge for the GLM simulation are composed of the observed discharge value of Grifone gauge and 38% of the simulated discharge representing that percentage of the catchment. Periods without observation data are either set to 0 m³s⁻¹ during dry seasons or filled based on expert knowledge considering rainfall data (24.01.2014 – 12.02.2014).

#### A3.2 Inflow water temperature

Observations of inflow water temperature exist only at Grifone gauge (see Fig. 3) for the periods 01.10.2015 – 13.05.2017 and 01.01.2017 – 06.07.2017. Values for periods with no observation data are calculated from the air temperature by subtracting 1°C from the daily mean value of air temperature. The computation by linear correlation was discarded, because the time series of water and air temperature show distinctive different relationships for the two applicable periods (with

constant stream runoff, Table A6).

**Table A6: Coefficient of correlation and linear relationship for water and air temperature for periods with constant runoff**

| Period | $R^2$ | Linear relationship |
|--------|-------|---------------------|
| 27.02.2016 – 13.05.2016 | 0.29 | y = 0.535x + 7.691 |
| 17.02.2017 – 11.05.2017 | 0.84 | y = 1.165x- 1.021 |

#### A3.2 Outflow discharge

The lake has no surface outflow, but an exfiltration to groundwater can be assumed. Niedda et al. (2014) estimate a seepage

value of 1.5 mm per day, which is represented by a constant outflow time series in the input data. The average lake surface area of 0.38 km² for the period are July 2011 to August 2017 is applied to calculate the outflow discharge value of 0.0066 ms⁻¹.





### A4 Lake field data

### A4.1 Lake level

Lake level data surveyed by a diver sensor place on the lake bottom (further details on the instrument setup see Giadrossich et al. (2015)) with an hourly resolution starting from 18.11.2011. Prior to this date manually measurements are available in a

weekly to biweekly resolution. Data from the diver are outputted as height (m) over the lake bottom. The position of the data logger at the lake bottom and thereby its elevation above sea level slightly changes with each re-installation after a data export on the surface. Hence, the data are corrected to refer all heights to the same elevation of 18.85 m a.s.l.. The data taken from the surface are also set to this reference elevation.

### A4.2 Lake water temperature

Water temperature data were observed automatically by the lake station and are available from 26.08.2012 measuring in the depths of 1, 2, 4, and 6m in hourly resolution. The surface and bottom water temperature were measured by a diver with an observation start on 26.07.2012 and 23.08.2012, respectively (for further details on the instrument setup see Giadrossich et al. (2015)). From 27.11.2015 also the depths of 3 and 5 m were observed. Prior to the automatically observations temperature profiles were collected manually every 2 to 5 weeks. Fig. A7 visualizes the available filed data of lake water

temperature. For the period of 24.09.2013 – 04.03.2014, when the lake station was not in operation (including the diver at the surface), the surface temperature is derived from the bottom temperature based on the assumption of isothermal conditions in the vertical lake profile, which is usual for the site in this season of the year.

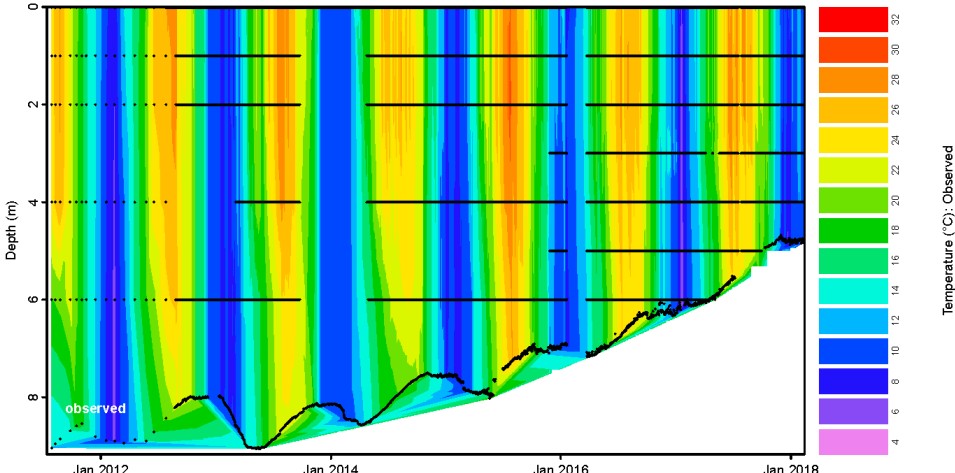

**Fig. A7: Visualization of available field data of water temperature (black dotts) and interpolated temperatures in the vertical**
**profile (glmGUI)**





### Appendix B: Input data for Lake Ammersee

### B1 Observation stations

The locations of hydrological gauging stations (discharge and groundwater) are displayed in Fig. 7. Meteorological observation stations are shown in Fig. B1. Lake field data are surveyed at the Lake station, where also meteorological

5  observation are taken. All data except cloud cover data (see chapter B2) are freely available at https://www.gkd.bayern.de/ provided by the Bavarian Environment Agency (Bay. LfU, 2018).

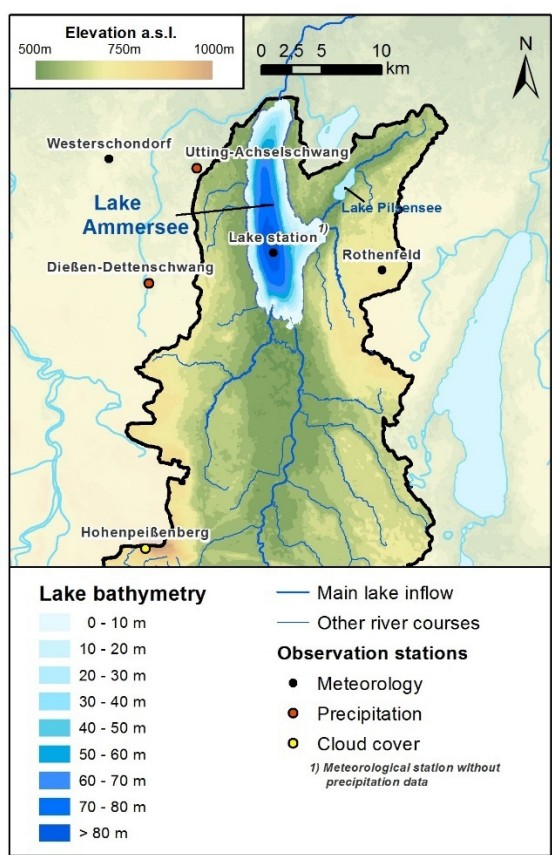

**Fig. B1: Bathymetry of Lake Ammersee and meteorological observation stations (Source DEM: Elevation data from ASTER GDEM, a product of METI and NASA, Source geo-data: Geobasisdaten © Bayerische Vermessungsverwaltung,**
10  **www.geodaten.bayern.de).**





### B2 Meteorological model input data

Observations of air temperature, wind speed, relative humidity, and shortwave radiation are taken from the lake station as meteorological input data for the simulation. The parameters are measured in a temporal resolution of 15 minutes. Shortwave radiation data are added to daily sums. For the other three parameters daily averages are calculated.

Missing values for wind speed, air temperature and relative humidity are calculated from observations (hourly values) of Rothenfeld station (Fig. B1). The latter two exhibit considerable seasonal variations in the difference of the measurements between the lake and Rothenfeld station (Fig. B2). Hence, missing values are computed by adding the average deviation for the simulation period of the respective month. The monthly mean values are computed based on a 19-day moving average. Missing data for wind speed are calculated from Rothefeld station observations by adding the mean offest value of the

simulation period of -1.3 ms$^{-1}$. Missing values of shortwave radiation data at the lake station are computed from the average of Rothefeld and Westerschondorf measurements (at both stations hourly observations) and no error correction further is required. Precipitation is inputted to the model as averaged values from observations of the stations Rothefeld, Utting-Achselschwang, and Dießen-Dettenschwang. The data are considered as snow for temperatures below 1°C as applied by Springer et al. (2015).

For Lake Ammersee the GLM option of using cloud cover instead of longwave radiation data is selected. Cloud cover data are nearest available at Hohenpeißenberg station (Fig. B1, operator: German Weather Service, free available at http://www.dwd.de/cdc, hourly observations) and daily averages are used without correction.





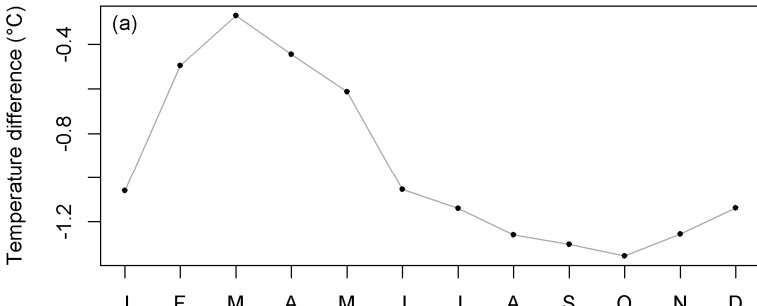

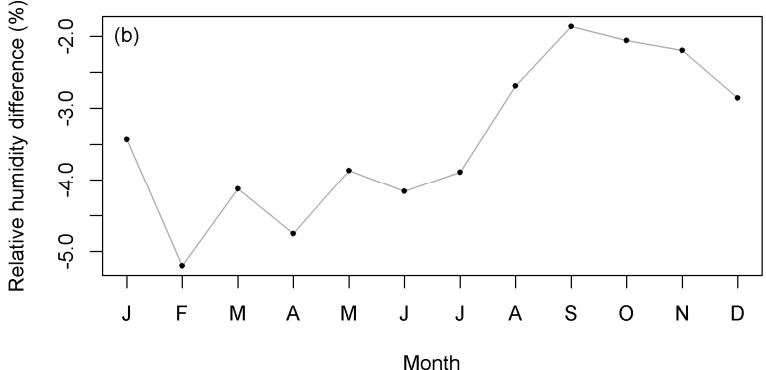

**Fig. B2:** Average difference (monthly mean of a 19-day moving average) between Rothenfeld and lake station of a) air temperature and b) relative humidity.





### B3 Hydrological model input data

#### B3.1 Inflow and outflow discharge

Stream discharge is observed only for the two major inflows River Ammer and River Rott plus the River Kienbach (Table B1). The hydrological contribution, concerning quantity and temporal distribution, of River Fischbach, of the other inflows

(all other smaller creeks summarized, see Fig. 7), and of the groundwater is unknown. To be able to reproduce the lake level applying GLM, four different inflows are defined. The time series represent the discharge data from 1. River Ammer, 2. River Fischbach, 3. groundwater inflow, and 4. the sum of River Rott, River Kienbach, and all other unknown inflows. Hereby, the real contribution of the respective inflow is approached by adjusting the inflow factors during the lake level calibration process.

Values of River Ammer time series are observed at Weilheim station (Fig. 7). The temporal pattern of Fischbach discharge is highly controlled by the outflow of Lake Pilsensee (Büche, 2018). The discharge of Fischbach is estimated by a simple rainfall-runoff relationship. The hydrograph (Fig. B3) is designed considering the retention of the lake showing a slow reaction of lake outflows on rainfall events (long smoothed decline of discharge) and the subsequent inflow to the water body (indicated by the quick rise of the hydrograph in the first days after the rainfall). With a runoff coefficient of 0.4 the

estimated discharge for the simulation period averages to 0.67 $m^3s^{-1}$. This mean value corresponds to the only hydrological value for Lake Pilsensee existing in the literature of 0.4 $m^3s^{-1}$ for the annual mean of the main inflow (Grimminger, 1982). Subsurface inflow to Lake Ammersee is estimated from groundwater level observation at Wielenbach (Fig. 7) calculated by a stage-discharge relation (Fig. B4). The inflow is set to insert the lake in a depth height of 79.15 m above the lake bottom, which is about 4 m below the surface (Bueche and Vetter, 2014a) dependent of the lake level.

The outflow of Lake Ammersee is observed at gauge station Stegen (Fig. 7) in a temporal resolution of 15 min. The discharge data are averaged and taken as outflow time series input data. After Kleinmann (1995) no subsurface outflows exist.

**Table B1: Characteristics of sub-catchments of Lake Ammersee. The data are representative for the stream inlet to the lake**

| (Sub)-Catchment | Gauge station | Catchment size (km²) | Percentage of total area (%) |
|---|---|---|---|
| Ammersee | Stegen | 994.6 | 100.0 |
| Other inflows | - | 122.0 | 12.3 |
| Fischbach | - | 56.2 | 5.6 |
| Kienbach | Herrsching | 12.4 | 1.2 |
| Rott | Raisting | 82.5 | 8.3 |
| Ammer | Weilheim | 721.5 | 72.5 |



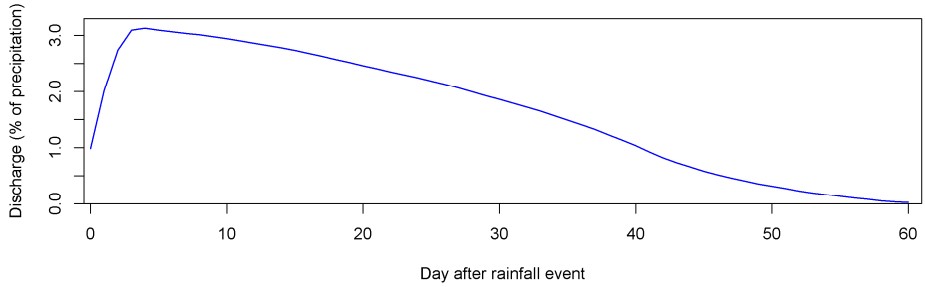

**Fig. B3: Hydrograph of River Fischbach for a rainfall event**

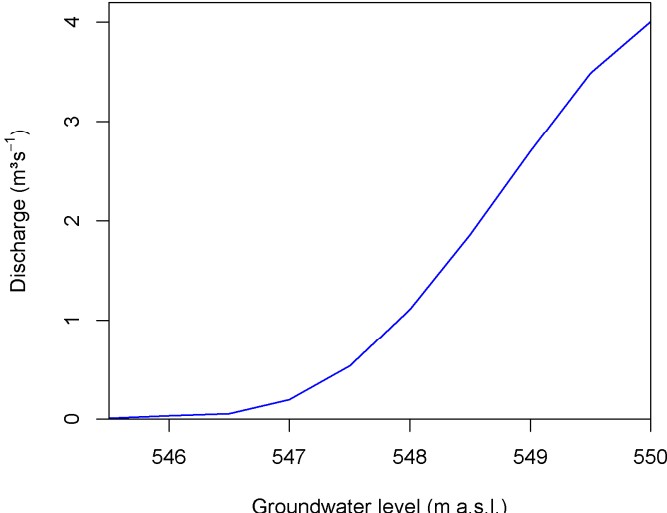

**Fig. B4: Stage-discharge relation for groundwater inflow to Lake Ammersee**

5    The discharge of Rott, Kienbach and all other unknown smaller creeks are defined as one discharge time series to the simulation. Missing values for Rott and Kienbach (maximum gap length = 9 days) are estimated by expert knowledge taking rainfall events into account. The discharge of the smaller inflows is calculated from Kienbach gauge data multiplying the values by the factor of 8.28 representing the relationship of the sub-catchment area sizes. This is feasible as Kienbach and the smaller inflows have similar characteristics in relief and runoff generation.



### B3.2 Inflow water temperature

Observations for inflow water temperatures are only available for River Ammer surveyed in hourly resolution (Gauge station Weilheim, Fig. 7) and values for water temperature are only specified for this surface inflow. However, this data can be seen as representative for the sum of all surface inflows. Water temperatures of groundwater can be estimated to be roughly

equivalent to the annual air temperature (Boehrer and Schultze, 2008) and a constant value of 8.65 °C derived from long-term average air temperature (Bueche and Vetter, 2014a) is applied for the subsurface inflow input data.

### B4 Lake filed data

The water temperatures of the lake are observed automatically by a lake buoy operated by the Bavarian Environment Agency. Data are surveyed in the depths (all in given in m) 0.5 (representing the surface), 1.0, 2.0, 4.0, 6.0, 8.0, 10.0, 12.0,

14.0, 16.0, 18.0, 20.0, 25.0, 40.0, 60.0, and 78.0 in a temporal resolution of 15 min, but already provided by the operator as daily averages. The available data are displayed in Fig. 2 (black dots) and these values are used as field data. Existing temperature values in May 2016 are evaluated as unrealistic high and excluded from the data set. Lake level data are observed at Stegen gauge station (Fig. 7) in a temporal resolution of 15 min, but also as daily means are provided. The data unit is elevation a.s.l.. To obtain lake level heights the values are subtracted by the elevation of the lake bottom (449.78 m

a.s.l.).

### Appendix C: Model input data and calibration parameter

**Table C1: Required model input data (after Hipsey et al., 2014)**

| Parameter | Unit | Description |
|---|---|---|
| Meteorological parameters: | | |
| Air temperature | °C | average air temperature 10 m above the water surface |
| Wind speed | $ms^{-1}$ | average wind speed 10 m above the water surface |
| Relative Humidity | % | average relative humidity (0 – 100 %) 10 m above the water surface |
| Shortwave radiation | $Wm^{-2}$ | average shortwave radiation |
| Longwave radiation | $Wm^{-2}$ | Longwave radiation input is assumed to be direct incident intensity |
| Rainfall | $md^{-1}$ | rainfall dept |
| Cloud cover *(optional)* | - | Required if no information on longwave radiation available and |



| | | incoming longwave flux is estimated from cloud cover fraction data |
|---|---|---|
| Snowfall *(optional)* | md⁻¹ | snowfall depth |
| **Hydrological parameters:** | | |
| Inflow discharge | m³s⁻¹ | Average discharge |
| Water temperature | °C | average streamflow water temperature |
| Salinity *(optional)* | mgl⁻¹ | streamflow salinity |
| Outflow discharge | m³s⁻¹ | average discharge |

**Table C2: List of preselected parameters for autocalobration (after Hipsey et al., 2017)**

| Parameter (glm.nml ID) | Description |
|---|---|
| **Lake Properties** | |
| Kw | Extinction coefficient for PAR radiation (unit: m⁻¹) |
| **Surface Dynamics** | |
| ch | Bulk aerodynamic coefficient for sensible heat transfer |
| ce | Bulk aerodynamic coefficient for latent heat transfer |
| cd | Bulk aerodynamic coefficient for transfer of momentum |
| **Mixing Parameters** | |
| coef_mix_conv | Mixing efficiency - convective overturn |
| coef_wind_stir | Mixing efficiency - wind stirring |
| coef_mix_shear | Mixing efficiency - shear production |
| coef_mix_turb | Mixing efficiency - unsteady turbulence (acceleration) |
| coef_mix_KH | Mixing efficiency - Kelvin-Helmholtz turbulent billows |
| coef_mix_hyp | Mixing efficiency of hypolimnetic turbulence |
| **Hydrological and meteorological factors** | |
| seepage_rate | Rate of seepage from the deepest layer (unit: m day⁻¹) |
| inflow_factor | Factor for inflow(s) |
| outflow_factor | Factor for outflow |

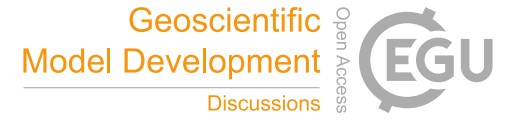

| Rain_factor | Factor for rainfall |
| wind_factor | Factor for wind speed |

**Appendix D: Contour plot of GLM simulations**

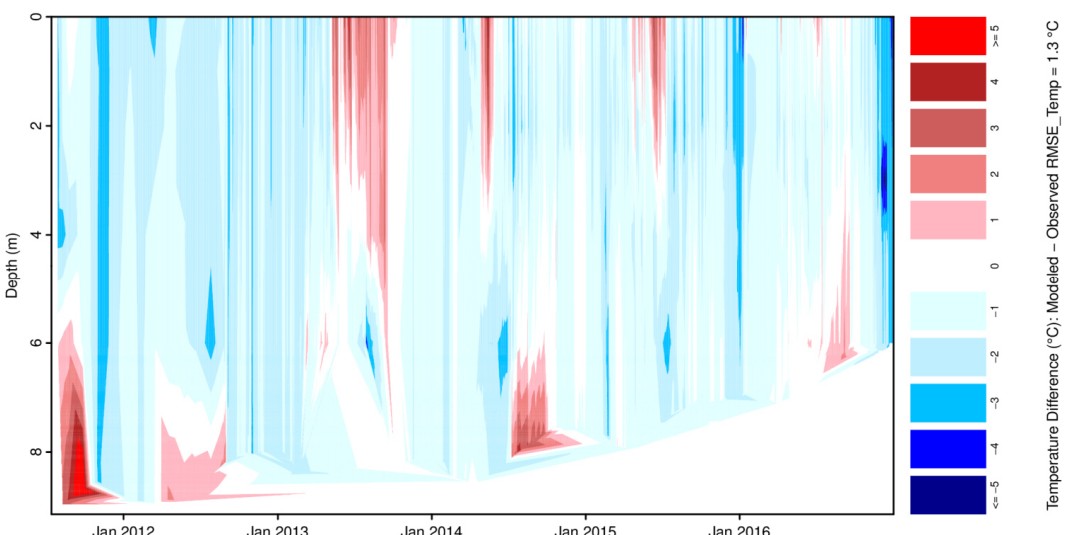

**Fig. D1: Contour plot of differences between simulated and observed water temperatures of Lake Baratz**

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
