# Peer review of "glmGUI v1.0: an R-based Graphical User Interface and toolbox for GLM (General Lake Model) simulations"

_Geoscientific Model Development, 2018_

## Referee Comment (RC1) · Andrea Fenocchi (Referee) · 14 Feb 2019

GENERAL COMMENTS: This paper introduces a new R-based graphical interface for the open-source General Lake Model (GLM), which simulates one-dimensional lake thermodynamics. It includes input and output processing and visualisation facilities, in addition to an autocalibration routine. I deem the creation of this GUI very useful, especially as it might introduce to lake thermodynamic modelling people who are not familiar with scripting. As I say this, I especially refer to governmental environmental agencies, who would strongly benefit from lake and reservoir thermodynamic modelling for their water management activities. In the manuscript, the created workspace

itself is clearly presented and the two validation case studies are suitable and properly developed, except from a few queries listed in the "Specific comments" section of this review. That's why I'm giving this paper the "minor revisions" judgement. However, the Authors should carefully inspect the manuscript, preferably with the help of an experienced (native) English speaker, as some sentences are unclear and there are many incorrect expressions. I highlighted a significant part of them in the "Technical corrections" list below, but there are many others to be addressed. Regarding the structure of the paper: 1) Appendices A and B are more suited to be included as part of the Supplementary material, forming two separate technical reports, given that the focus of the paper and of the journal is on the glmGUI package and not on the development of the case studies; 2) Appendices C and D should be moved to the main text, the former because having to look up each time many pages forward in the paper to understand the meaning of model parameters makes understanding passages troublesome for people unfamiliar with GLM, the latter because the counterpart figure for Lake Ammersee (Fig. 9) is already part of the main text. The paper would strongly benefit from reorganising the material in the Appendices, receiving a more compact outlook.

SPECIFIC COMMENTS: P4, L10-11. Use of a Kalman filter to fill missing values of meteorological time series should be briefly discussed, highlighting the advantages with respect to simpler interpolation methods. P4, L15-16. This is true for shallow lakes, such as Lake Baratz, but doesn't hold for lakes already as deep as Lake Ammersee, at least for ordinary level variations. Please revise. P5, L1-2. While a single RMSE water temperature error indicator may be significant for shallow lakes, in which however the 1D approximation is less reliable, for deep holomictic lakes it is usually better to distinguish the errors for surface and deep waters, as a single metric would be strongly biased by hypolimnetic temperatures displaying small variations from the initial conditions, i.e. small errors, hiding larger errors in the epilimnion. I think that the possibility to consider multiple metrics across the lake depth and also relative error ones (e.g. rRMSE) should be introduced in a future update of the package. Please discuss this. P5, L3. Please define the MBE index. P5, L15-16. Please explain better. P5, L17-18. This sentence is not needed. P6, L6-7. Reverse "input" and "output" in the sentence. A sensitivity analysis investigates the sensitivity of the model results with respect to variations of the input parameters, not the opposite. P7, L1-2. Please explain better. P7, L21-22. The "r" parameter should be the size of the sample (i.e. the number of tested values) for each calibrated parameter, not the resolution (i.e. the spacing between tested values). P8, L5. What is the Grifone station being used for? It'not clear at this point in the manuscript. P9, L12-14. What did Lenhart et al. (2002) state? Please explain better. P9, L12-14. The Authors should disclose that wind affects the simulation of lake levels through its influence on evaporation. P9, L18-22. This passage should be explained better. P10, Fig. 4. Such information would be more efficiently conveyed by a table (see Table 1 for the Lake Ammersee case). P15, L12. What is the meaning of "factors of discharge"? P15, L20-22. What was the total number of performed simulations and how long did they take overall? I would like to ask the same also for the previous Lake Baratz case. P16, L4-5. The RMSE and MBE values reduced with respect to what? P16, L11-12. This contradicts the statement at P16, L6-8. Please clarify. P16, Fig. 8. Looking at this plot I would not state that the achieved model fit is "very satisfactory" (P16, L5). Large errors dominate for most of the simulated period. P18, L6-9. The Authors should discuss the possibility to employ in the future more refined calibration methods than the adopted plain Monte Carlo approach, such as MCMC (Markov chain Monte Carlo) and other methods, which allow better addressing the computational effort. P18, L14-16. Please explain better. P18, L27-28. The Authors should stress that the main benefit of GLM in a hydrological analysis is that lake evaporation is calculated with higher accuracy than by using classic formulas. P20, L12. Why and how was the observation setup demolished? It's just my personal curiosity. P20, L17. I do not understand well the meaning of the $R^2$ index for the lake station itself. Please explain. P21, L6-8. Specify that these are average differences. P23, L7-8. Please explain better.

TECHNICAL CORRECTIONS: P1, L29-30. The structure of the sentence is twisted. Write something like: "Lakes are often considered to be "sentinels of change", as they

respond to changes in their environment (Williamson et al., 2009, Hipsey et al., 2017)".
P2, L6-9. This sentence is too long and should be split. P2, L11. Do the Authors
mean "reservoirs" in place of "irrogations"? P2, L14. Add a comma after "GLM". P2,
L17. Do the Authors mean "goal" in place of "maxim"? P2, L19. Replace "usability
friendly" with "user-friendly". P3, L2. Don't capitalize "Mediterranean", it might induce
confusion in the name of the lake. P3, L2-4. This sentence should be revised. P3,
L8-9. This sentence should be revised. P3, L18. Replace "in and" with "at a". P3,
L22. Remove "of" after "R", replace "the" with "a" after "and". P3, L25. Add a hyphen
between "window" and "based". P4, L5-6. Replace "control" with "input". P4, L7-8.
This sentence should be revised. P5, L3-4. This sentence should be revised. P5,
L9-11. This sentence should be revised. P6, L2-3. Black dots cannot be seen, as
they make up black lines by being aligned and manifold. P8, L12. This sentence
should be revised. P9, L2-5. This sentence should be revised. P9, L18. The sentence
"According..." should begin in a new line. P10, L10. Replace "proofs" with "proves".
P11, L5-8. These sentences should be revised. P13, L3. Replace "is" with "of". P13,
L4. Replace "seasons" with "years". P15, L5. Add a comma after "31.12.2017". P15,
L10. Capitalise "Secchi-disk". P15, L12. Add a colon after "inflows" in place of "of".
P16, L6-9. These sentences should be revised. P17, L4. Replace "in" with "at" after
"epilimnion". P17, L5. Replace "satisfactory" with "satisfactorily". P17, L12. Do the
Authors mean "cornerstone" in place of "centerpeace"? P17, L16. Replace "for" with
"by". P18, L1-3. This sentence should be revised. P18, L4-5. This sentence should
be revised. P18, L18. Remove "the". P18, L23. Do the Authors mean "efficiency" in
place of "expedience"? P18, L25. Replace "smaller" with "shallower". P18, L32-33.
This sentence should be revised. P19, L3. Replace "of" with "for". P19, L9. Remove
"expenditure of". P19, L21. Add "this" after "in". P20, L10. Replace "environment" with
"proximity", "a" with "an", "rafting on" with "located at". P20, L16. Replace "station" with
"stations". P22, L6-8. This sentence should be revised P23, L7. Replace "form" with
"from". P25, L9-10. The structure of the sentence is twisted. P29, L3. Add "were"
before "surveyed", replace "place" with "placed". P29, L4. Replace "manually" with

"manual". P30, L5. Replace "observation" with "observations". P31, L3. Replace "in" with "at". P31, L5. Replace "of" with "at". P31, L16. Replace "free" with "freely". P35, L2. Replace "in" with "at". P35, L12. Replace "unrealistic" with "unrealistically". P35, L13. Replace "in" with "at".

---

## Short Comment (SC1) · 12 Mar 2019

Dear Dr. Fenocchi,
many thanks for your helpful comments on our paper. I can only refer to the points that relate to my area of expertise within this study, which consists mainly of half of the programming work and minor contributions to the text of the paper.

The first specific comment I'm referring to is the following: "P4, L10-11. Use of a Kalman filter to fill missing values of meteorological time series should be briefly discussed, highlighting the advantages with respect to simpler interpolation methods."

[Figure]

For the interpolation of meteorological time series I have tested several different methods, e.g. linear interpolation, spline interpolation, moving average with simple, linear and exponential weighting, last-observation-carried-forward (LOCF) and the Kalman smoothing. I have tested these methods by manually removing values from existing time series of meteorological data and interpolating these missing values. The Kalman smoothing could deliver the best and most constant results for all kinds of meteorological data within this test and was therefore implemented in the package.

The second (and last) specific comment I'm referring to is the following: "P7, L21-22. The "r" parameter should be the size of the sample (i.e.the number of tested values) for each calibrated parameter, not the resolution (i.e. thespacing between tested values)."

Yes, you're absolutely right; that was misleading. We will change the explanation and formula and express it like this:

"The runtime of the calibration algorithm ($t_{cal}$) increases exponentially with the number of parameters ($p$) to be calibrated
(Eq.(1))
$$t_{cal} = r^p \cdot (t_{GLM} + t_{RMSE}),$$
with $r$ as the number of tested values for each parameter $p$ and $t_{GLM}$, $t_{RMSE}$ as runtimes of the lake model and the calculation of the output RMSE."

---

## Referee Comment (RC2) · Anonymous Referee #2 · 18 May 2019

See attached file

Please also note the supplement to this comment:
https://www.geosci-model-dev-discuss.net/gmd-2018-314/gmd-2018-314-RC2-supplement.pdf
* * *

---

## Author Comment (AC1) · 14 Jun 2019

The comment was uploaded in the form of a supplement:
https://www.geosci-model-dev-discuss.net/gmd-2018-314/gmd-2018-314-AC1-supplement.pdf

---

## Author Comment (AC2) · 14 Jun 2019

**Author Response to reviewer 2:**

We are grateful for the comments and address the following:

| Reviewer's comment | Authors' comment | Suggestion of changes in the manuscript |
|---|---|---|
| 1.  "Use of a Kalman filter to fill missing values of meteorological time series should be briefly discussed" | Already answered: https://editor.copernicus.org/index.php/gmd-2018-314-SC1.pdf?_mdl=msover_md&_jrl=365&_lcm=oc108lcm109w&_acm=get_comm_file&_ms=73376&c=157993&salt=1095167195190759503 | |
| 2.  P7, L1-2. Please explain better | This is addressed in responses to reviewer 1 | |
| 3.  P8, L5. What is the Grifone station being used for? It'not clear at this point in the manuscript | Yes, no need to mention Grifone here, so just remove the short sentence at P8 L5. It is already written at P28 L4.
Grifone is the place where data are taken (discharge and water temperature) at a stream gauge. Please check for Fig. 3 and Appendix A. | Removal of the short sentence at P8 L5 |
| 4.  P9, L12-14. What did Lenhart et al. (2002) state? Please explain better. | This is addressed in responses to reviewer 1 | |
| 5.  P9, L12-14. The Authors should disclose that wind affects the simulation of lake levels through its influence on evaporation. | Thank you for this remark. The addition of this information makes the choice of the parameter for the sensitivity analysis clearer. | We will add this remark and rewrite this passage the following (P9, L11):
" ... wind_factor as wind can impact the lake level due to its influence on evaporation. The option ..." |
| 6.  P9, L18-22. This passage should be explained better. | | P9, L19:
"... considering the **three** parameters ..."
P9, L22. "... calibration. Not considering parameter ch is plausible, as its SI value matches only just the threshold to be medium sensitive." |

| | | |
|---|---|---|
| 7.      P10, Fig. 4. Such information would be more efficiently conveyed by a table (see Table 1 for the Lake Ammersee case). | We think a figure depicts the information easier and more intuitive than a table. Hence, we would remain this figure here. | |
| 8.       P15, L12. What is the meaning of "factors of discharge"? | It is meant the *inflow_factor.* | We will revise: "*... considers the inflow_factor (i.e. simple factor for the discharge values of the inflows) ...* " |
| 9.      P15, L20-22. What was the total number of performed simulations and how long did they take overall? I would like to ask the same also for the previous Lake Baratz case. | We will add information on the number of simulation and calibration runs. The simulation time of autocalibration runs varied from several hours to 3 days dependent from the number of calibration parameters (please see also the response to reviewer 1, comment 1) | For Lake Baratz, we will add at P9, L22: "*Within the calibration process in total approx. 3000 simulation runs in 6 calibration runs were conducted.*" For Lake Ammersee, we will add at the end of the section 4.2 on (P15): "*In total approx. 50000 simulation runs in 12 calibration runs were conducted within the calibration process*". |
| 10.      P16, L4-5. The RMSE and MBE values reduced with respect to what? | The RMSE and MBE values are reduced by the calibration of the inflow factors instead of "default" inflow factors of 1.0 for all tributaries. | We change the sentence as follows: "*By using these adapted inflow factors instead of the default value of 1.0, the overall RMSE reduced significantly from 1.10 mto 0.20 m, and the MBE from - 1.00 m to 0.09 m, and the achieved model fit can be assessed as very satisfactory.*" |
| 11.      P16, L11-12. This contradicts the statement at P16, L6-8. Please clarify. | Correct, this might be too optimistic. We will remove the latter statement | Removal of the sentence P18, L18/19 |
| 12.      P16, Fig. 8. Looking at this plot I would not state that the achieved model fit is "very satisfactory" (P16, L5). Large errors dominate for most of the simulated period. | Indeed, there are differences, but the general, natural behaviour of the curve is following the general conditions. | |
| 13.      P18, L6-9. The Authors should discuss the possibility to employ in the future more refined calibration methods than the adopted plain Monte Carlo approach, such as MCMC | The reviewer is absolutely correct. Other calibration methods could help reduce computational resources and time. As this is the initial version of the package, this | |

| | | |
|---|---|---|
| (Markov chain Monte Carlo) and other methods, which allow better addressing the computational effort. | will be improved in upcoming versions using more efficient calibration methods. | |
| 14.       P18, L14-16. Please explain better. | Due to the calibration algorithm of percental alterations, values of parameters are created (outputted) with a high decimal precision. For example: Applying an autocalibration for an initial value of 0.23 (= default value of coef_wind_stir) with a range of 30% and for an interval of 5, RMSE will be calculate for parameter values of 0.299, 0.2645, 0.23, 0.1955, 0.2093. In our opinion the explanation is clearly formulated | |
| 15.       P18, L27-28. The Authors should stress that the main benefit of GLM in a hydrological analysis is that lake evaporation is calculated with higher accuracy than by using classic formulas. | The GLM uses the commonly adopted bulk aerodynamicformula to estimate the latent heat flux and therefore evaporation (Hipsey et al., 2014). The bulk formula is **the** classic method to estimate evaporation over homogenous areas (for instance lakes or oceans; Foken 2006, 128). If the "hydrological analysis" just uses e.g. Penman-Monteith for the whole catchment (including the lake), it is a major upgrade for the estimation of evaporation to use the GLM/Bulk formula for the lake area.
 Therefore we propose adding the sentence passage stated right to the manuscript: | *"The GLM uses the bulk aerodynamic formula to estimate the latent heat flux and therefore evaporation (Hipsey et al., 2014), which is commonly applied to assess the evaporation rate over open water bodies (Fischer et al., 1979; Hicks, 1972). Including the GLM in the hydrological analysis can therefore improve the accuracy of the modeled evaporation and thus the water balance estimate."* |
| 16.       P20, L12. Why and how was the observation setup demolished? It's just my personal curiosity. | Majority of the land around the lake is private, and the University of Sassari had an agreement with an owner for many years. However, once the agreement expired it was not possible to renew it and the station was dismantled. | |

| | | |
|---|---|---|
| 17. P20, L17. I do not understand well the meaning of the $R^2$ index for the lake station itself. Please explain. | Yes, this is misleading and the information is obsolete, please see also responses to reviewer 1 | Removal of the brackets and their content |
| 18. P21, L6-8. Specify that these are average differences.

19. P23, L7-8. Please explain better. | OK

It makes no sense to obtain by calculation values exceeding the maximum observed value. Hence the maximum possible value is fixed to be 97,0, but stated wrong in the manuscript | Specification will be added

Replacement of 96,7 by 97,0 |

---

## Author Response (AR1)

**Point-by-point Response to reviewer 1:**

Dear Andrea,

Thank you very much for the very detailed and helpful review. In our response we will address to your suggestions on the manuscript structure (GENERAL COMMENTS) and your SPECIFIC COMMENTS. Regarding your TECHNICAL COMMENTS the manuscript will be checked again by a professional editor considering your comments. Please note, that some comments and revisions in the manuscript has changed since our Final Response in the Discussion forum. Please also apologize mixing up some comments with the other reviewer in our Final Response.

Changes in the manuscript are highlighted in this document by blue color. **Pages and lines of our changes in the manuscript are referring to the marked up version of it.**

| Reviewer's comment | Authors' comment | Suggestion of changes in the manuscript |
|---|---|---|
| GENERAL COMMENTS | | |
| 1. Appendices A and B are more suited to be included as part of the Supplementary material, forming two separate technical reports, given that the focus of the paper and of the journal is on the glmGUI package and not on the development of the case studies. | Your assessment is very valuable for us and will follow you suggestions | We convert the Appendices A and B into a Supplemental Materials |
| 2. Appendices C and D should be moved to the main text, the former be-cause having to look up each time many pages forward in the paper to understand the meaning of model parameters makes understanding passages troublesome for people unfamiliar with GLM, the latter because the counterpart figure for Lake Ammersee | We wanted to avoid overloading the manuscript with large tables. Figure D1 is erroneously the same as Fig. 9, it was supposed to be the same plot type as already presented, but f or the simulation results of Lake Ammersee. As you stated above the manuscript's focus should remain on the glmGUI, the presentation of an second plot of | |

| | | |
|---|---|---|
| (Fig.9) is already part of the main text. The paper would strongly benefit from reorganizing the material in the Appendices, receiving a more compact outlook. | the same type in the main part would not have a great benefit. | |

| SPECIAL COMMENTS | | |
|---|---|---|
| 1.  P4, L10-11. Use of a Kalman filter to fill missing values of meteorological time series should be briefly discussed" | Already answered: https://editor.copernicus.org/index.php/gmd-2018-314-SC1.pdf?_mdl=msover_md&_jrl=365&_lcm=oc108lcm109w&_acm=get_comm_file&_ms=73376&c=157993&salt=1095167195190759503 | We add a short description on the test of the filter method (P4, L11): "*Several filter methods were tested by manually removing values from existing time series of meteorological data and interpolating these missing values. The Kalman smoothing could deliver the best and most constant results for all kinds of the various meteorological data.*" |
| 2.  P4, L15-16.  This is true for shallow lakes, such as Lake Baratz, but doesn't hold for lakes already as deep as Lake Ammersee, at least for ordinary level variations.  Please revise. | The referee is right. We add a remark considering that the impact of the lake level is true for shallow lakes | We revise the sentence as the following (P5, L2/3): "*As lake level variations can have a strong impact on the water temperature distribution within lake (especially true for shallow waters), the validation*" |
| 3.  P5, L1-2.  While a single RMSE water temperature error indicator may be significant for shallow lakes, in which however the 1D approximation is less reliable, for deep holomictic lakes it is usually better to distinguish the errors for surface and deep waters, as a single metric would be strongly biased by hypolimnetic temperatures displaying small variations from the initial conditions, i.e.  small errors, hiding larger errors in the epilimnion.  I think that the possibility to consider multiple metrics across the lake depth and also relative | This is correct, but the identification of the epilimnion and hypolimnion layer is complex and different for every lake. An automatic computation by the GUI, which would have to be very generic to be applicable to a wide range of water bodies, might yield errors due to the generalized approach. Hence, we think that such kind of postprocessing has to be done by the user in a subsequent step. But a more detailed examination of the spatial distribution of the model deviations is already provided by the GUI by the output option of | |

| | | |
|---|---|---|
| error ones (e.g. rRMSE) should be introduced in a future update of the package. Please discuss this. | difference plots.
As suggested in the comment, the implementation of further metrics shall be provided in upcoming versions of the package. | |
| 4.  P5, L3. Please define the MBE index. | | We add after the abbreviation MBE within the brackets (P5, L6):
", average of the lake level differences of all time simulated time steps" |
| 5.  P5, L15-16. Please explain better | | We add at the end of the sentence (P5, L20):
" ...but lake level variations are represented by changes in depth, *which become visible at the "bottom" of this plot type*." |
| 6.  P5, L17-18. This sentence is not needed. | This part was weak, which was also remarked by reviewer 2. We revise and describe in more detail for a better understanding of the changes and to underline of the improvement of the plot settings in contrast to the existing of *glmtools*. | P5 L22:
*"The default settings to scale the color bar legend for water temperature plots taking into account the range of temperatures and also erroneously the range of lake depth. This method is adopted in glmGUI, but discarding the consideration of the lake depth, and the temperature range is adjusted explicitly to the plotting method to provide well differentiated color ranges in the legend."* |

| 7. P6, L6-7. Reverse "input" and "output" in the sentence. A sensitivity analysis investigates the sensitivity of the model results with respect to variations of the input parameters, not the opposite. | Thank you. We revise as suggested | (P6, L6): *"It investigates how variations in the output of a numerical model of can be attributed to variations of input parameters or factors"* |
|---|---|---|
| 8. P7, L1-2. Please explain better | We should have added the formula and the selection options for the sampling of the parameter space. | We add the following at P7, L1: "The widely used approach after Lenhart et al. (2002) is implemented in the GUI. *The Sensitivity Index (SI) is calculated for each selected parameter separately, as only one parameter is changed at a time.* $$SI = \frac{(y_1 - y_2)/y_0}{2\Delta x/x_0}$$ *The parameter with the value $x_0$ is increased and decreased by $\Delta x$. The resulting outputs $y_1$ and $y_2$ (either water temperature, lake level or the respective RMSEs) are subtracted and normalized by the output $y_0$, which results from using the unchanged parameter value $x_0$. $\Delta x$ can be set to four different values in the GUI (5%, 10%, 20%, 50%).*" |
| 9. P7, L21-22.The "r" parameter should be the size of the sample (i.e.the number of tested values) foreach calibrated parameter, not the resolution (i.e. thespacing between tested values). | Already answered in our Specific Comment: https://editor.copernicus.org/index.php/gmd-2018-314-SC1.pdf?_mdl=msover_md&_jrl=365&_lcm=oc108lcm109w&_acm=get_comm_file&_ms=73376&c=157993&salt=1095167195190759503 | We adjust the equation and its description: " $$t_{cal} = r^p * (t_{GLM} + t_{RMSE})$$ *with r as the number of tested values for each parameter p and …*" |

| | | |
|---|---|---|
| 10. P8, L5. What is the Grifone station being used for? It'not clear at this point in the manuscript | Yes, no need to mention Grifone here, so just remove the short sentence at P8 L5. It is already written at P28 L4.
Grifone is the place where data are taken (discharge and water temperature) at a stream gauge. Please check for Fig. 3 and Appendix A. | Removal of the short sentence at P8 L11 |
| 11. P9, L12-14. What did Lenhart et al. (2002) state? Please explain better. | Please see comment 6: we addressed to this by more details of the Sensitivity Index | |
| 12. P9, L12-14. The Authors should disclose that wind affects the simulation of lake levels through its influence on evaporation. | Thank you for this remark. The addition of this information makes the choice of the parameter for the sensitivity analysis clearer. | We will add this remark and rewrite this passage the following (P10, L7):
" ... *wind_factor as wind can have an impact on the lake level due to its influence on evaporation. The option ...*" |
| 13. P9, L18-22. This passage should be explained better. | | P10, L16:
"... *considering these* **three** *parameters ...*"
P10, L19. "... *calibration. Not considering parameter ch is plausible, as its SI value matches only just the threshold to be medium sensitive.*" |
| 14. P10, Fig. 4. Such information would be more efficiently conveyed by a table (see Table 1 for the Lake Ammersee case). | We think a figure depicts the information easier and more intuitive than a table. Hence, we would remain this figure here. | |
| 15. P15, L12. What is the meaning of "factors of discharge"? | It is meant the *inflow_factor.* | We will revise (P16, L12):
"... *considers the inflow_factor (i.e. simple factor for the discharge values of the inflows) ...*" |
| 16. P15, L20-22. What was the total number of performed simulations and how long did they take overall? I would like to ask the same also for the previous Lake Baratz case. | We will add information on the number of simulation and calibration runs. The simulation time of autocalibration runs varied from several hours to 3 days dependent from the number of calibration parameters (please see also the response to reviewer 2, comment 1) | For Lake Baratz, we will add at P10, L20:
"*Within the calibration process a total number of approx. 3000 simulation runs were conducted in 6 autocalibration runs*"
For Lake Ammersee, we will add at the end of the section 4.2 on (P16, L24): "*In total approx.* |

| | | *50000 simulation runs in 12 autocalibration runs were performed within the calibration process”.* |
|---|---|---|
| 17.  P16, L4-5. The RMSE and MBE values reduced with respect to what? | The RMSE and MBE values are reduced by the calibration of the inflow factors instead of "default" inflow factors of 1.0 for all tributaries. | We change the sentence as follows (P17, L4): *"By using these adapted inflow factors instead of the default value of 1.0, the overall RMSE reduced significantly from 1.10 m to 0.20 m, and the MBE from - 1.00 m to 0.09 m, and the achieved model fit can be assessed as very satisfactory."* |
| 18.  P16, L11-12. This contradicts the statement at P16, L6-8. Please clarify. | Correct, this might be too optimistic. We will remove the latter statement | Removal of the sentence P18, L18/19 |
| 19.  P16, Fig. 8. Looking at this plot I would not state that the achieved model fit is "very satisfactory" (P16, L5). Large errors dominate for most of the simulated period. | Indeed, there are differences, but the general, natural behaviour of the curve is following the general conditions. | |
| 20.  P18, L6-9. The Authors should discuss the possibility to employ in the future more refined calibration methods than the adopted plain Monte Carlo approach, such as MCMC (Markov chain Monte Carlo) and other methods, which allow better addressing the computational effort. | The reviewer is absolutely correct. Other calibration methods could help reduce computational resources and time. As this is the initial version of the package, this will be improved in upcoming versions using more efficient calibration methods. | |
| 21.  P18, L14-16. Please explain better. | Due to the calibration algorithm of percental alterations, values of parameters are created (outputted) with a high decimal precision. For example: Applying an autocalibration for an initial value of 0.23 (= default value of coef_wind_stir) with a range of 30% and for an interval of 5, RMSE will be calculate for parameter values of 0.299, 0.2645, 0.23, 0.1955, 0.2093. In our opinion the explanation is clearly formulated | |

| | | |
|---|---|---|
| 22. P18, L27-28. The Authors should stress that the main benefit of GLM in a hydrological analysis is that lake evaporation is calculated with higher accuracy than by using classic formulas. | The GLM uses the commonly adopted bulk aerodynamic formula to estimate the latent heat flux and therefore evaporation (Hipsey et al., 2014). The bulk formula is **the** classic method to estimate evaporation over homogenous areas (for instance lakes or oceans; Foken 2006, 128). If the "hydrological analysis" just uses e.g. Penman-Monteith for the whole catchment (including the lake), it is a major upgrade for the estimation of evaporation to use the GLM/Bulk formula for the lake area. Therefore we propose adding the sentence passage stated right to the manuscript: | We add (P20, L23): "*The GLM uses the bulk aerodynamic formula to estimate the latent heat flux and therefore evaporation (Hipsey et al., 2014), which is commonly applied to assess the evaporation rate over open water bodies (Fischer et al., 1979; Hicks, 1972). Including the GLM in the hydrological analysis can therefore improve the accuracy of the modeled evaporation and thus the water balance estimate.*" |
| 23. P20, L12. Why and how was the observation setup demolished? It's just my personal curiosity. | Majority of the land around the lake is private, and the University of Sassari had an agreement with an owner for many years. However, once the agreement expired it was not possible to renew it and the station was dismantled. | |
| 24. P20, L17. I do not understand well the meaning of the R^2 index for the lake station itself. Please explain. | Yes, this is misleading and the information is obsolete, please see also responses to reviewer 2 | Removal of the brackets and their content (P22, L22) |
| 25. P21, L6-8. Specify that these are average differences. | OK | Specification added (Supp. Mat, P2, L7) |
| 26. P23, L7-8. Please explain better. | It makes no sense to obtain by calculation values exceeding the maximum observed value. Hence the maximum possible value is fixed to be 97,0, but stated wrong in the manuscript | Replacement of 96,7 by 97,0 |

**Point-by-point Response to reviewer 2:**

Dear reviewer,

We are grateful for the comments and address the following:
(Please note, that some comments and revisions in the manuscript has changed since our Final Response in the Discussion forum. Please also apologize mixing up some comments with the other reviewer in our Final Response).

Changes in the manuscript are highlighted in this document by blue color. **Pages and lines of our changes in the manuscript are referring to the marked up version of it.**

| Reviewer's comment | Authors' comment | Suggestion of changes in the manuscript |
|---|---|---|
| 1. Autocalibration routine- this is perhaps one of the most value features in the GUI as the calibration process can be difficult and time consuming. However, no information is given on how the calibration is actually performed, what are the objective functions, how is the parameter space sampled, what are the stopping criteria, how flexible is the routine to user definitions, and other issues. Furthermore, how good is the calibration tool in relation to manual calibration? I would like to see a much more extensive description, testing and discussion of the calibration process. | Example: If the user chooses to calibrate the model for the two parameters P1 (default value 1.0) and P2 (default value 2.0), she/he has to choose the upper and lower calibration limit by a percentage range. Let's say she/he wants to vary P1 by 10% only and P2 by 50%. → P1 will be in the range of 0.9 to 1.1 and P2 between 1.0 and 3.0. Then the user chooses the resolution of the space between these ranges. Let's say the user wants to have only 4 samples in the percentage range plus the default value. So P1 will get the following values: 0.9, 0.95, 1.0, 1.05, 1.1 P2 gets: 1.0, 1.5, 2.0, 2.5, 3.0

Now all possible combinations of P1 and P2 are tested: 5 x 5 combinations So the GLM.exe is run 25 times by the glmGUI and the resulting RMSE of lake level and water temperature is saved to two CSV-files. So the | BEispiele von der Disc einbaueb.

We revised the sentence pg. 20. ln. 4: *"At present running the tool on a server in advance or as a "background task" can compensate this problem"* |

| | user can see, which combination results in which RMSE. | |
| --- | --- | --- |
| | This process is already described in the manuscript as follows (pg. 7, ln. 16): "*The user can choose out of these parameters those which are to be included in the calibration process and define a percentage range, by which the upper and lower limit of every parameter is changed from the value in the glm2.nml-file. The resolution of the increase/decrease of the parameters within the defined limits can be set as well. According to these settings, model runs of GLM are executed with all possible combinations of the selected parameters ("brute-force"). The overall RMSE of the lake level or water temperature is calculated and saved for every parameter combination to a csv file, so the "best fit" is indicated.* " | |

- Objective function is RMSE (as described)
- Sampling of the parameter space is described
- There is no stopping criterium, as all possible combinations are calculated(as described)
- The flexibility is given by selecting:  1) The set of parameters
2) The percentual ranges of upper/lower limits for each parameter
3) The resolution of the increase/decrease of the parameters

| | | |
|---|---|---|
| | within the defined limits

Furthermore, saving the RMSE of **all parameter combinations** gives the experienced user the possibility to choose not the "best fit" parameter set, if the user thinks that one parameter of this set might have an unrealistic value.

The comparison to a manual calibration is very subjective, as it depends on the user and her/his skill. We can just state, that the autocalibration function can simplify the calibration process, as the user does not have to re-run the GLM.exe for each parameter change manually. Additionally, using this tool can save time, e.g. when running on a server in advance or as a "background task". | |
| 2. Sensitivity analysis- This too is very useful however there is insufficient information on how it is actually conducted. How is SI calculated? Is the analysis conducted by changing one parameter (or variable) at a time or changing all at the same time?
How is the parameter space sampled for the analysis? While some of the meteorological variables are included in the SA I would also expect shortwave and longwave radiation to be included as they can be difficult to measure accurately, especially the later. | Yes, we totally agree on that. We should have added the formula and the selection options for the sampling of the parameter space. | We add the following at pg. 7 ln.17:
"*The widely used approach after Lenhart et al. (2002) is implemented in the GUI. The Sensitivity Index (SI) is calculated for each selected parameter separately, as only one parameter is changed at a time:*

$SI = \dfrac{(y_1 - y_2)/y_0}{2\Delta x / x_0}$

*The parameter with the value $x_0$ is increased and decreased by $\Delta x$. The resulting outputs $y_1$ and $y_2$ (either water temperature, lake level or the respective RMSEs) are subtracted and normalized by the output $y_0$, which results from using the unchanged parameter value $x_0$.*
*$\Delta x$ can be set to four different values in the GUI* |

| | | |
|---|---|---|
| | | *(5%, 10%, 20%, 50%).*" |
| 3. Along these lines, including quantifiable indices for the goodness of fit of the model to lake-based data is critical and the authors have included RMSE and MBE. I think the authors should include a range of indices which the user can select from when conducting the analysis. | Adding more quantifiable indices is relatively easy to implement in future Releases (V1.1). | |
| 4. In the Lake Baratz lake level results (fig. 5) there is a period during which the fit between the model and lake data is not good in contrast to the other periods. I would like to see discussion of this and possible explanations. Similarly for Lake Ammersee. You mention the issue in lines 10-13 (pg 16) but don't attempt to explain the discrepancy. I think the large discrepancies that are obvious on Fig. 8 need to be explained. | For Lake Baratz (Fig. 5) | We add to description of these results in section 3.3 the following at pg 12, ln 4): "*A period of remarkable deviation of simulated and observed lake level is induced in January of 2014, which could be attributed to uncertainties in the discharge input data simulated by a hydrological model. The basin in that period of the year is still in an intermediate status of soil moisture. Probably the hydrological model overestimated the discharge on the base of rainfall events in January 2014 which in reality did not produce a significant lake level variation (see Fig. 2 in Giadrossich et al., 2015). However, the observed and simulated lake levels in Fig. 5 have the same trend and the error caused in January is propagated throughout the year.*" |
| | For Lake Ammersee (Fig. 8): | We insert to the presentation of the results (Section 4.3) pg17 ,L11: "*No obvious explanation for these trend shifts could be found, although a detailed investigation of the existing hydrological data was conducted. An impact of a highly complex* |

| | | |
|---|---|---|
| | | *groundwater inflow system is likely to have a key role in the water balance of the lake, which is not considered by the applied input data sufficiently. Furthermore it cannot be ruled out that unknown alterations or errors in the observation setup of the gauges cause these "turning points" as some of them correspond to flood events, which might have implied problems with the measurements.*" |
| 5. Conclusions section- I think this section requires significant strengthening in order to better convey the key points. The way it is currently written does not touch on allthe important points and mentions issues that are not necessary. | We addressing already to plenty key points of within the conclusion:
- enhancement of R tools by the presented package and its benefits
- the test of the package for two example (sites) with a comprehensive description of the input data
- the applicability of the package regarding test of input data quality (new)
- decent simulation results
- the advantage of the open-source provision, especially for the opportunity of a usage for integrative investigations (e.g coupling with ecological models) (expanded)
- outlook for further applications
We are convinced, that this is a well structured, precise, and sufficient conclusion. | |
| 6. Input data for Lake Baratz- you mention a 5 month gap in met data (pg 20 lines12-13) how did you deal with this gap? | The entire preprocessing of the meteorological input including the dealing of the observation gap of the lake station is described in detail for each parameter in Section A2. | To clarify we add an indication after line 13 (pg 22):
"*A detailed description of the source and required processing steps of the respective parameters is given in the section A2.*" |
| Fig. A1- the air temperature data from Fertila | The data at Fertilia station were observed with | We add to section A2.1 (pg22, ln26) "*Values at* |

| | | |
|---|---|---|
| station does not look like continuous data. What type of data were these? | a precision of 1 degree, which cause the distribution in the graph. | *Fertilia station were available in a precision of 1 degree*". |
| Fig. A6-
Isn't it possible that the unique water transparency event in 2017 affected the relationship shown in this figure and that a different equation is required for that period? Please discuss. | We rephrased the paragraph form line 1 to line 5 at page 9 because it was not clear. There is not a unique water transparency event in 2017 that affected the relationship. The sensitivity of Kw is low for the whole period and doesn't change significantly, giving an average light extinction coefficient value Kw = 0.57. Thus, we considered 0.57 to be representative of the whole period. It has been obtained dividing the Secchi-disk constant ranging from a minimum value of 1.44 to a maximum of 1.80 (Hornung, 2002; Holmes, 1975; Chapra, 1997) divided by Secchi-disk depth ranging from 2.50 to 3 meters. In these cases, the Kw values range between 0.48 and 0.72. The value of 0.57 has been adopted, because lake had a higher depth I the period between 13.07.2011 to 31.12.2016. If we would apply the constant = 1.44 and secchi-depth=2.5, and constant 1.7 (as suggested by Poole and Atkins, 1929), secchi-depth = 3, we will obtain the same value of 0.57. | We add an exhaustive description pg 9, ln: "*The simulation period for Lake Baratz is determined to be 13.07.2011 to 31.12.2016. We assume the light extinction coefficient value Kw = 0.57 is representative of the whole study period. Kw is calculated dividing the Secchi-disk constant (in this case the minimum value of 1.44 was taken as it usually ranges between 1.44 to 1.80, Hornung, 2002; Holmes, 1975; Chapra, 1997), by the mean Secchi-disk depth of 2.50 meters (data from June 2016 to June 2017). A similar value can be obtained considering a Secchi-disk average depth of 3 meters (assumed when the lake had a higher water level) and Secchi-disk constant of 1.70 (Poole and Atkins, 1929).*"

New REFEENCES:
Holmes, R. W.: The secchi disc depth in turbid coastal water. Limnology and Oceanography 15, 688–694,1975.

Chapra, S. C.: Surface Water-Quality Modeling, international edn. McGraw-Hill, 1997.

Hornung, R.: Numerical Modelling of Stratification in Lake Constance with the 1-D hydrodynamic model DYRESM, http://dx.doi.org/10.18419/opus-163, 2002 |

| | | Poole, H. H. and Atkins,W. R. G., Photo-electric measurements of submarine illumination throughout the year.Journal of the Marine Biological Association of theUnited Kingdom,16,297–324, 1929. |
|---|---|---|
| 7. Section A4.2 field data- in the main text you mention that mixing occurs in the winter however here you state that you assume isothermal conditions from 24.09.13. Do isothermal conditions develop as early as September? | Isothermal conditions establish usually during early autumn. In this year the lake had already a very small temperature gradient before the station was out of action. | We will add at the end of the paragraph (now Supp. Mat., pg. 10, line 14): "*…profile. Homothermy is usual for the site in this season of the year and the vertical temperature gradient of 0.91° C on 24.09.2014 was already low indicating no stable thermal stratification.*" |
| 8. English- the MS needs to be edited by a native English speaker or professional editor. Currently there are many sections/sentences that need rewriting. | The entire manuscript is checked again by a professional editor. | |
| 9. The shutdown button in the GUI is in German and not English. Better to have it in English like the rest of the GUI. | Button label depends on language settings of R. One solution could be to name the button with an unique string. gbutton ("Close window", …) | |
| 10. Pg 5 ln 18: erroneously - what do you mean? | | We will revise the sentence(s) (pg. 5, ln. 22): "*…water temperature plots taking into account the range of temperatures and also erroneously the range of lake depth. This method is adopted in glmGUI, but discarding the consideration of the lake depth, and the temperature range…*" |
| 11. pg 13 ln 6- outflow or inflow? | Yes, we are talking about the outflow | We reformulate (pg. 13, ln. 6): "*The lake has a catchment area of about 994 km² and its outflow in the north (Stegen gauge station).*" |
| 12. Pg 16 ln 4- the RMSE reduced significantly- | The reason for the reduction of the RMSE is due | |

| | | |
|---|---|---|
| under which conditions? Why? | to the application of the above mentioned inflow factors. | |
| 13. Fig 9- isn;t the lake 83 m deep? If so, why is only 9 m shown? | Thank you for your meticulous review! This is erroneously the Figure for Lake Baratz and will be replace by the right plot for Lake Ammersee | The figure shows erroneously results for Lake Baratz and will be replaced by the plot for Lake Ammersee |
| 14. Pg 19 ln11- "This includes a data quality assessment..."- That is not the case. The GUI allows visualization but does not include, as far as I understand, QA tools. | | We reformulate here (pg. 19, ln. 11):" *The GUI includes tools to check the quality of the input data. This comprises the option of a visual detection of errors, missing values and plausibility.*" |
| 15. Pg 19 ln 15- sentence not clear. | | We rewrite this sentence in the following manner: Pg. 21, ln. 11:"*The GUI allows a high level of interoperability due to the option of combining with other operating systems.*" *Pg 21, ln. 20: "Furthermore, we designed the software with the aim of a high flexibility for the application of other scenarios, for various study areas or with diverse time steps."* |
| 16. Pg 20 ln 17- R2 between which two sets of data? | The information in brackets is misleading here and can just be removed | Information in brackets are removed |
| 17. Pg 22 ln 7- why correct only data after 21.6.2016 and not the earlier data if they are much lower than Fertilia station | Fertilia station is quite in a distance to the lake and the data measured at the lake station before June 2016 are reliable, which is confirmed by observations at the closer Grifone station (systematically lower for the available period until 2014). Hence only data after June 2016 were corrected. | We will add sentence at (now in Supp. Mat) pg. 2, ln 8: "*Observations at Grifone station are in the range of the measurements taken at the lake station before 21.06.2016 and hence, these data were taken as reference for correction.*" |

[revised manuscript text omitted]

**Technical Report: Input data for GLM simulations of Lake Baratz and Lake Ammersee**

**S1 Input data for Lake Baratz**

**S1.1 Observation stations**

Meteorological data are taken from several stations in the environment of the lake including an observation station rafting on the lake surface center. Hydrological data (discharge and water temperature) were surveyed at a stream gauge at the Grifone site, where the observation setup was demolished on 31.05.2017. **Fehler! Verweisquelle konnte nicht gefunden werden.** (main paper) gives an overview of the locations of the stations. The lake station was not in operation from 24.09.2013 to 25.04.2014 (Giadrossich et al., 2015). A detailed description of the source and required processing steps of the respective parameters is given in the section S1.2.

**S1.2 Meteorological model input data**

**S1.2.1 Air temperature**

Input data for air temperature are taken from the respective station in the following order:

- Lake station (R² > 0.93, reference period: 08.07.2011 – 23.09.2013, Giadrossich et al., 2015)
- Calculated from Grifone station by linear regression of the lake station ($R^2 = 0.97$, reference period: 25.04.2014 – 31.05.2017, Fig. S1a)
- Calculated from Fertilia station by linear regression of the lake station ($R^2 = 0.99$, reference period: 15.01.2015 – 12.02.2018, Fig. S1b). Values at Fertilia station were available in a precision of 1 degree.

[Figure]

**Fig. S1: Linear correlation of air temperature for the lake station and a) Grifone station and b) Fertilia station.**

**S1.2.2 Wind speed**

Wind speed data at the lake station have several observations gaps and the measurements show a significant bias between the periods before and after 21.06.2016 (Fig. S2), when a new sensor was installed after an outage. The bias is detected by comparing to data obtained at Fertilia station. For the period before this date the average wind speeds measured at the lake station were 1.19 ms$^{-1}$ (average difference) lower than measured at Fertilia station. After the 21.06.2016 the average difference was only 0.08 ms$^{-1}$. Observations at Grifone station are within the range of the measurements taken at the lake station before 21.06.2016, hence these data were taken as reference for correction.

[revised manuscript text omitted]

**S1.4 Lake field data**

**S1.4.1 Lake level**

Lake level data surveyed by means of a diver sensor placed on the lake bottom (further details on the instrument setup see Giadrossich et al. (2015)) with an hourly resolution starting from 18.11.2011. Prior to this date manually measurements

5   are available in a weekly to biweekly resolution. Data from the diver are outputted as height (m) over the lake bottom. The position of the data logger at the lake bottom and thereby its elevation above sea level slightly changes with each re-installation after a data export on the surface. Hence, the data are corrected to refer all heights to the same elevation of 18.85 m a.s.l.. The data taken from the surface are also set to this reference elevation.

**S1.4.2 Lake water temperature**

10   Water temperature data were observed automatically by the lake station and are available from 26.08.2012 measuring in the depths of 1, 2, 4, and 6m in hourly resolution. The surface and bottom water temperature were measured by a diver with an observation start on 26.07.2012 and 23.08.2012, respectively (for further details on the instrument setup see Giadrossich et al. (2015)). From 27.11.2015 also the depths of 3 and 5 m were studied. Prior to the automatically observations temperature profiles were collected manually every 2 to 5 weeks. Fig. S7 visualizes the available filed data of

15   lake water temperature. For the period of 24.09.2013 – 04.03.2014, when the lake station was not in operation (including the diver at the surface), the surface temperature is derived from the bottom temperature based on the assumption of isothermal conditions in the vertical lake profile. Homothermy, is common for the site in this season of the year. and the vertical temperature gradient of 0.91° C on 24.09.2014 was already low indicating no stable thermal stratification.

[Figure]

**Fig. S7: Visualization of available field data of water temperature (black dots) and interpolated temperatures in the vertical profile (glmGUI).**

**S2 Input data for Lake Ammersee**

**S2.1 Observation stations**

The locations of hydrological gauging stations (discharge and groundwater) are displayed in **Fehler! Verweisquelle konnte nicht gefunden werden.** in the main text. Meteorological observation stations are shown in (Fig. S8). Lake field data are surveyed at the Lake station, where also meteorological observations are taken. All data except for cloud cover data (see chapter B2) are freely available at https://www.gkd.bayern.de/ provided by the Bavarian Environment Agency (Bay. LfU, 2018).

[Figure]

**Fig. S8: Bathymetry of Lake Ammersee and meteorological observation stations (Source DEM: Elevation data from ASTER GDEM, a product of METI and NASA, Source geo-data: Geobasisdaten © Bayerische Vermessungsverwaltung, www.geodaten.bayern.de).**

**S2.2 Meteorological model input data**

Observations of air temperature, wind speed, relative humidity, and shortwave radiation are gathered at the lake station as meteorological input data for the simulation. The parameters are measured in a temporal resolution of 15 minutes. Shortwave radiation data are added to daily sums. For the other three parameters daily averages are calculated.

10  Missing values for wind speed, air temperature and relative humidity are calculated from observations (hourly values) of Rothenfeld station (Fig. S8). The latter two are subject to considerable seasonal variations in the difference of the

measurements between the lake and Rothenfeld station (Fig. S8). Hence, missing values are computed by adding the average deviation for the simulation period of the respective month. The monthly mean values are computed based on a 19-day moving average. Missing data for wind speed are calculated from Rothefeld station observations by adding the mean offest value of the simulation period of -1.3 ms$^{-1}$. Missing values of shortwave radiation data at the lake station are

5   computed from the average of Rothefeld and Westerschondorf measurements (at both stations hourly observations) and no further error correction is required. Precipitation is inputted to the model as averaged values from observations of the stations Rothefeld, Utting-Achselschwang, and Dießen-Dettenschwang. According to Springer et al. (2015) , the data are recognized as snow as soon as temperatures are below 1°C.

For Lake Ammersee the GLM option of using cloud cover instead of longwave radiation data is selected. The closest

10   location for cloud cover data is Hohenpeißenberg station (Fig. S8, operator: German Weather Service, free available at http://www.dwd.de/cdc, hourly observations) and daily averages are used without correction.

[Figure]

[Figure]

**Fig. S9: Average difference (monthly mean of a 19-day moving average) between Rothenfeld and lake station of a) air temperature and b) relative humidity.**

**S2.3 Hydrological model input data**

**S2.3.1 Inflow and outflow discharge**

Stream discharge is observed only for the two major inflows River Ammer and River Rott plus the River Kienbach (Table S7). The hydrological contribution, concerning quantity and temporal distribution, of River Fischbach, of the other inflows (all other smaller creeks summarized, see **Fehler! Verweisquelle konnte nicht gefunden werden.** in the main text), and of the groundwater is unknown. To be able to reproduce the lake level applying GLM, four different inflows are defined. The time series represent the discharge data from 1. River Ammer, 2. River Fischbach, 3. groundwater inflow, and 4. the sum of River Rott, River Kienbach, and all other unknown inflows. Hereby, the real contribution of the respective inflow is approached by adjusting the inflow factors during the lake level calibration process.

Values of River Ammer time series are observed at Weilheim station (**Fehler! Verweisquelle konnte nicht gefunden werden.**, main paper). The temporal pattern of Fischbach discharge is highly controlled by the outflow of Lake Pilsensee (Büche, 2018). The discharge of Fischbach is estimated by a simple rainfall-runoff relationship. The hydrograph (Fig. S10) is designed considering the retention of the lake showing a slow reaction of lake outflows on rainfall events (long smoothed decline of discharge) and the subsequent inflow to the water body (indicated by the quick rise of the hydrograph in the first days after the rainfall). With a runoff coefficient of 0.4 the estimated discharge for the simulation period averages to 0.67 $m^3s^{-1}$. This mean value corresponds to the only hydrological value for Lake Pilsensee existing in the literature of 0.4 $m^3s^{-1}$ for the annual mean of the main inflow (Grimminger, 1982).

Subsurface inflow to Lake Ammersee is estimated from groundwater level observation at Wielenbach (**Fehler! Verweisquelle konnte nicht gefunden werden.**, main paper) calculated by a stage-discharge relation (Fig. S11). The inflow is set to insert the lake at a depth of 79.15 m above the lake bottom, which is about 4 m below the surface (Bueche and Vetter, 2014) dependent on the lake level.

The outflow of Lake Ammersee is observed at gauge station Stegen (**Fehler! Verweisquelle konnte nicht gefunden werden.**, main paper) in a temporal resolution of 15 min. The discharge data are averaged and taken as outflow time series input data. After Kleinmann (1995) no subsurface outflows exist.

[revised manuscript text omitted]

---

## Author Response (AR3)

**Author's Response**

Dear reviewers and editors,

as suggested we carefully checked the manuscript and technical report and found a few typographical errors, which we corrected. Please see below our responses to the comments of referee #3 in blue. Additionally we added the source of the DEM depicted in Figure 3.

We corrected the number of the manuscript section "discussion" from 5 to 6.

In the .zip folder containing the figures, we added also the figures of the Technical Report, which is part of the Supplementary Material. As they are numbered with Sxx we named them fSxx.

Kind Regards

Thomas Bueche, Mark Vetter, Benjamin Poschlod, Marko Wenk, Filippo Giadrossich, Mario Pirastru

Kind Regards

Thomas Bueche

Minor typos.
GUI normally means graphical user interface. In several places this is referred to as a Geographical user interface - i.e. in title, line 15 page 1, line 19 page 2 whereas in rest of text the more common graphical user interface is used. Which did you actually mean?
Thank you for this important comment. We use 'graphical' now in all expressions.

I am confused what the EGU guidelines are for using italics. In Table A1 and Table the various parameters are NOT in italics. In the text they are usually in italics, but not always. I'll highlight the places they are not in italics in text.

Page 8 line 5 p not in italics.
Page 9 line 8 KW not in italics.
Page 10 line 18 ch not in italics.
Page 16 line 13 Inflow_factor not italicized
We turned the identified terms to italic style.

Awkward wording:

Page 14 line 9 - Error message for missing reference - "Error! Reference source could not be found."
Corrected

Line 19 page one. Change distinguishing to distinguished.
In our opinion distinguishing is the correct term here.

Line 27 "self written extension" might be changed. Who wrote it? The computer? Or the authors?
We rephrased: "… written by the authors interacts…"

Line 23 page 5. Change to "The second plot…" done

Page 21 Line 5. "Wraps up" usually means "finished" or "ends", whereas I think you mean "includes"

or "combines" here. Thank you for the suggestion, we used "combines" instead "wraps up"

Page 21 line 10. Rather than "close the gap" I think you meant "fills the gap" done

Page 22. Line 9. I think you meant "funding" rather than "founding". If you dd mean "founding" it would only make sense to say "founding a one month Professorship for.."
Thank you for the comment, we changed to "funding"

[revised manuscript text omitted]

**Technical Report: Input data for GLM simulations of Lake Baratz and Lake Ammersee**

**S1 Input data for Lake Baratz**

**S1.1 Observation stations**

Meteorological data are taken from several stations in the environment of the lake including an observation station rafting on the lake surface center. Hydrological data (discharge and water temperature) were surveyed at a stream gauge at the Grifone site, where the observation setup was demolished on 31.05.2017. Fig. 3 (main paper) gives an overview of the locations of the stations. The lake station was not in operation from 24.09.2013 to 25.04.2014 (Giadrossich et al., 2015). A detailed description of the source and required processing steps of the respective parameters is given in the section S1.2.

**S1.2 Meteorological model input data**

**S1.2.1 Air temperature**

Input data for air temperature are taken from the respective station in the following order:

- Lake station Calculated from Grifone station by linear regression of the lake station ($R^2 = 0.97$, reference period: 25.04.2014 – 31.05.2017, Fig. S1a)
- Calculated from Fertilia station by linear regression of the lake station ($R^2 = 0.99$, reference period: 15.01.2015 – 12.02.2018, Fig. S1b). Values at Fertilia station were available in a precision of 1 degree.

[Figure]

**Fig. S1: Linear correlation of air temperature for the lake station and a) Grifone station and b) Fertilia station.**

**S1.2.2 Wind speed**

Wind speed data at the lake station have several observations gaps and the measurements show a significant bias between the periods before and after 21.06.2016 (Fig. S2), when a new sensor was installed after an outage. The bias is detected by comparing to data obtained at Fertilia station. For the period before this date the average wind

speeds measured at the lake station were 1.19 ms$^{-1}$ (average difference) lower than measured at Fertilia station. After the 21.06.2016 the average difference was only 0.08 ms$^{-1}$. Observations at Grifone station are within the range of the measurements taken at the lake station before 21.06.2016, hence these data were taken as reference for correction.

[revised manuscript text omitted]

**S1.4 Lake field data**

**S1.4.1 Lake level**

Lake level data surveyed by means of a diver sensor placed on the lake bottom (further details on the instrument setup see Giadrossich et al. (2015)) with an hourly resolution starting from 18.11.2011. Prior to this date manually measurements are available in a weekly to biweekly resolution. Data from the diver are outputted as height (m) over the lake bottom. The position of the data logger at the lake bottom and thereby its elevation above sea level slightly changes with each re-installation after a data export on the surface. Hence, the data are corrected to refer all heights to the same elevation of 18.85 m a.s.l.. The data taken from the surface are also set to this reference elevation.

**S1.4.2 Lake water temperature**

Water temperature data were observed automatically by the lake station and are available from 26.08.2012 measuring in the depths of 1, 2, 4, and 6 m in hourly resolution. The surface and bottom water temperature were measured by a diver with an observation start on 26.07.2012 and 23.08.2012, respectively (for further details on the instrument setup see Giadrossich et al. (2015)). From 27.11.2015 also the depths of 3 and 5 m were studied. Prior to the automatically observations temperature profiles were collected manually every 2 to 5 weeks. Fig. S7Fig. S7 visualizes the available filed data of lake water temperature. For the period of 24.09.2013 – 04.03.2014, when the lake station was not in operation (including the diver at the surface), the surface temperature is derived from the bottom temperature based on the assumption of isothermal conditions in the vertical lake profile. Homothermy is common for the site in this season of the year and the vertical temperature gradient of 0.91 °C on 24.09.2014 was already low indicating no stable thermal stratification.

[Figure]

**Fig. S7: Visualization of available field data of water temperature (black dots) and interpolated temperatures in the vertical profile (glmGUI).**

**S2 Input data for Lake Ammersee**

**S2.1 Observation stations**

The locations of hydrological gauging stations (discharge and groundwater) are displayed in Fig. 7 in the main text. Meteorological observation stations are shown in (Fig. S8). Lake field data are surveyed at the Lake station, where also meteorological observations are taken. All data except for cloud cover data (see chapter S.2.2B2) are freely available at https://www.gkd.bayern.de/ provided by the Bavarian Environment Agency (Bay. LfU, 2018).

[Figure]

**Fig. S8: Bathymetry of Lake Ammersee and meteorological observation stations (Source DEM: Elevation data from ASTER GDEM, a product of METI and NASA, Source geo-data: Geobasisdaten © Bayerische Vermessungsverwaltung, www.geodaten.bayern.de).**

**S2.2 Meteorological model input data**

Observations of air temperature, wind speed, relative humidity, and shortwave radiation are gathered at the lake station as meteorological input data for the simulation. The parameters are measured in a temporal resolution of 15 minutes. Shortwave radiation data are added to daily sums. For the other three parameters daily averages are calculated.

Missing values for wind speed, air temperature and relative humidity are calculated from observations (hourly values) of Rothenfeld station (Fig. S8). The latter two are subject to considerable seasonal variations in the difference of the measurements between the lake and Rothenfeld station (Fig. S98). Hence, missing values are computed by adding the average deviation for the simulation period of the respective month. The monthly mean values are computed based on a 19-day moving average. Missing data for wind speed are calculated from Rothefeld station observations by adding the mean offest value of the simulation period of -1.3 ms$^{-1}$. Missing values of shortwave radiation data at the lake station are computed from the average of Rothefeld and Westerschondorf measurements (at both stations hourly observations) and no further error correction is required. Precipitation is inputted to the model as averaged values from observations of the stations Rothefeld, Utting-

Achselschwang, and Dießen-Dettenschwang. According to Springer et al. (2015), the data are recognized as snow as soon as temperatures are below 1.0 °C.

[revised manuscript text omitted]